# Spectrally-Partitioned Flow Matching for Medical Time Series

## Abstract

This work introduces **Spectrally-Partitioned Flow Matching**, a novel and lightweight generative paradigm designed to address the spectral shortcomings of existing models in augmenting medical time series (TS) datasets. We propose an architecture that decouples signal generation into two specialized, jointly-trained components: a **Structure** model which learns to generate the core low-frequency (LF) waveforms, and a **Detail** model that conditionally generates the high-frequency (HF) details. Our approach prioritizes accessibility, requiring minimal VRAM capacity and leveraging a simple, fast ODE solver to generate high-fidelity, conditional multivariate data (e.g., ECG and EEG). We validate the quality of our synthetic data across various datasets through standard benchmarks and its utility through a downstream task, confirming that our formulation produces novel samples that correctly capture the full frequency spectrum without sacrificing quality or diversity.

## 1. Introduction

In machine learning, **generative modeling** (GM) has emerged as a central research topic, as it gives us a deeper understanding of complex data distributions by learning the joint distribution $p(\mathbf{x}, \mathbf{c})$ between input features $\mathbf{x}$ and their corresponding target variables $\mathbf{c}$. This capability extends beyond prediction via the conditional probability distribution modeled by discriminative models, $p(\mathbf{c}|\mathbf{x})$, allowing for the creation of new, synthetic samples that reflect the underlying data-generating process (Goodfellow et al., 2014).

The benefits of GM are particularly pronounced in domains where data acquisition is costly, labor-intensive, or limited by privacy and regulatory constraints. In medicine, for example, the need to protect patient confidentiality and adhere

to strict data-sharing protocols often restricts access to large scale datasets, posing challenges for training data-hungry deep learning models. Generative models offer a practical solution by enabling the creation of realistic synthetic medical data, which can be used to augment insufficient datasets, support collaborative research, and facilitate algorithm validation without compromising sensitive information (Chen et al., 2021).

While promising, applying GM to complex, multivariate medical TS like Electrocardiogram (ECG) and Electroencephalogram (EEG) data presents a unique challenge. These signals are composed of distinct components: long-range, low-frequency waveforms (like baseline rhythm) and rapid, high-frequency local events (like a QRS complex or a sleep spindle). Standard generative models often struggle to capture these properties simultaneously, resulting in *blurry* (over-aggregated) signals or missing critical *spiky* details, associated with the HF components. Furthermore, biomedical signals typically exhibit a $1/f$ power spectrum, where low-frequency components possess significantly higher amplitude than high-frequency details. This creates a gradient dominance problem: the loss function is overwhelmingly driven by large-amplitude structural errors, effectively drowning out the gradients required to learn fine-grained textures.

To address this, we introduce **Spectrally-Partitioned Flow Matching (SPFM)**, a novel framework. SPFM separates the generative process based on spectral properties, using two specialized, jointly-trained Conditional Flow Matching (CFM) models: a **Structure** model learns the LF global context conditioned on class labels $y$: $p(\mathbf{x}_{\text{LF}}|y)$ and a **Detail** model conditionally generates the HF local events, given both the class label and the generated LF-structure: $p(\mathbf{x}_{\text{HF}}|\mathbf{x}_{\text{LF}}, y)$. By disentangling the frequency components, we are effectively normalizing the learning signals, preventing the structure from starving the details of gradient updates and resulting in a more spectrally-balanced framework, an effect we coin **gradient democracy** (see Appendix B.2.1). Furthermore, by modularizing the training scheme, we achieve a lightweight high-fidelity generative dual-model framework, making it usable on standard hardware without sacrificing performance.

[1]Anonymous Institution, Anonymous City, Anonymous Region, Anonymous Country. Correspondence to: Anonymous Author <anon.email@domain.com>.

Preliminary work. Under review by the International Conference on Machine Learning (ICML). Do not distribute.

Our contributions are summarized as follows:

- We introduce **SPFM**, a flexible framework that disentangles time series generation into **Structure** (LF) and **Detail** (HF) components.

- We validate its effectiveness by generating high-fidelity, conditional data and provide a comprehensive evaluation using a set of statistical metrics, comparing our performance against computationally heavier baselines.

- We compare different backbone architectures as well as loss combinations across four **multivariate ECG and EEG datasets**, in order to show the flexibility and adaptability of our framework.

- We demonstrate the practical utility of our framework in the downstream task of **Spectral Super-Resolution** often referred to as bandwidth extension in signal processing (Kuleshov et al., 2017).

## 2. Theory of Generative Modeling

Given a dataset $D_{train} = \{\mathbf{x}_0, \cdots, \mathbf{x}_n\}$ sampled from an intractable distribution $p_{data} \subset \mathbb{R}^{d \times L}$, deep generative models (DGMs) aim to approximate $p_{data}$ via a parametric generator $g_\theta : \mathcal{Z} \to \mathcal{X}$. The core objective is to learn a mapping where samples from a tractable prior distribution $\mathcal{Z}$ (typically Gaussian) are transformed into valid data points:

$$\hat{\mathbf{x}} = g_\theta(\mathbf{z}), \quad \text{where} \quad \mathbf{z} \sim \mathcal{N}(0, \mathbf{I}) \tag{1}$$

If the model is expressive enough and $\theta$ is appropriately optimized, $g_\theta$ transports the simple noise density into the complex data density $p_{data}$.

A generator is conditioned if it is defined as $g_\theta | \mathcal{Y} \times \mathcal{Z} \to \mathcal{X}$., mapping from the product of the latent space $\mathcal{Z}$ and another space $\mathcal{Y}$ (e.g. discrete labels). The training objective becomes for the synthetic samples $\hat{\mathbf{x}}$ generated by $g_\theta$ to match the conditional distribution, using Bayes' rule: $p(\mathbf{x}|\mathbf{y}) = \frac{p(\mathbf{y}|\mathbf{x})}{p(\mathbf{y})}$. Conditioning improves flexibility and sample quality (Bao et al., 2022), and sampling techniques such as classifier- and classifier-free guidance (CFG) achieve state of the art (SoA) results in a number of different modalities (Dhariwal & Nichol, 2021).

### 2.1. Backbone Architectures for Sequence Modeling

The functional transformation $g_\theta$ described in 1 is usually parameterized by a deep neural network. The choice of backbone architecture is critical, as it dictates the model's ability to capture temporal dependencies and its computational efficiency. Over the past decade, the **Transformer** architecture has established itself as the standard for sequence modeling (Vaswani et al., 2017). Central to its success is the self-attention mechanism, which allows the model to route information between any two positions in a sequence effectively. However, the computational and memory complexity scales quadratically with sequence length ($\mathcal{O}(L^2)$), making them computationally prohibitive for the long-context TS data often encountered in medical applications, where the length can easily exceed thousands of timesteps.

To address the inefficiency of Transformers, **Structured State Space Models (SSMs)**, such as **S4** (Gu et al., 2022), have emerged as a powerful alternative. SSMs map a 1D input sequence $\mathbf{x}(t)$ to an output $\mathbf{y}(t)$ through a latent state $h(t)$ using a continuous-time linear differential equation:

$$h'(t) = \mathbf{A}h(t) + \mathbf{B}x(t), \quad y(t) = \mathbf{C}h(t) \tag{2}$$

Where $\mathbf{A}$, $\mathbf{B}$, and $\mathbf{C}$ are the state, input and output matrices respectively. Discretized for deep learning, these models achieve linear scaling ($\mathcal{O}(L)$) and can be computed efficiently via parallel scans or Fast Fourier Transforms (FFTs). However, standard SSMs rely on time-invariant parameters, which limits their ability to perform content-based reasoning or selectively propagate information based on the input context.

The **Mamba** architecture (Gu & Dao, 2023) represents the modern evolution of the SSM paradigm. It introduces a selection mechanism that makes the model's parameters input-dependent. This allows the model to selectively remember or ignore information at each timestep, quite similarly to the gating mechanisms in LSTMs, while retaining the parallel training efficiency of CNNs and the linear inference cost of SSMs.

### 2.2. Spectral Shortcomings of Generative Models

A pervasive challenge in GM, particularly for high-dimensional TS, is the inability to adequately reproduce high-frequency fidelity. Neural networks exhibit a phenomenon known as **spectral bias** (Rahaman et al., 2019), wherein the model learns low-frequency components significantly faster than high-frequency components. Consequently, generated signals often appear *oversmoothed* or lacking the stochastic sharpness inherent in real-world biological data, especially in TS domains, where data typically follows a $1/f$ power law (see Appendix B), meaning there is a vast energy disparity between structural components and transient events.

In standard single-model architectures such as Diffusion-TS or TimeGrad (Rasul et al., 2021), optimization is driven by a global objective like the Mean Squared Error (MSE), which inherently leads to **gradient dominance**: the high-amplitude low-frequency errors dominate the gradient signal, forcing the optimizer to prioritize the global trend while

neglecting high-frequency details. While backbone architectures like PatchTST (Nie et al., 2023) attempt to mitigate this via temporal segmentation, they still largely rely on global loss functions that do not explicitly account for this spectral disparity. This forces the easiest optimization path to be chosen at the expense of spectral quality, thus resulting in gradient starvation for the HF components (Pezeshki et al., 2021).

Furthermore, generating time series samples in FM and diffusion models entails solving differential equations via numerical solvers (e.g., Euler, Midpoint) that approximate the generative trajectory. This process exhibits a behavior analogous to the Nyquist-Shannon sampling theorem: the numerical solver acts as a **low-pass filter** (Shannon, 1949). To accurately reconstruct high-frequency oscillations in the data, the solver must utilize step sizes smaller than the characteristic time-scale required to resolve those frequencies. If the step size is too large, these rapid temporal dynamics are effectively averaged out, and while the generated signals may look globally coherent, they can be clinically invalid in a medical context.

## 3. Related Work

In the broader machine learning context, there have been several new GM families that frame the generative process with a differential equation (either ordinary or stochastic). **Diffusion Models** (Yang et al., 2022), train a network to progressively remove Gaussian noise added to data through a forward diffusion process, learning to reverse this to recover samples from noise. Formally, they model

$$p_\theta(\mathbf{x}_0) := \int p_\theta(\mathbf{x}_{0:T}) d\mathbf{x}_{1:T},$$

where $\mathbf{x}_1, ..., \mathbf{x}_T$ are latent states, and the reverse process $p_\theta(\mathbf{x}_{0:T})$ is a Markov chain with learned Gaussian transitions. **Diffusion-TS**, presented in (Yuan & Qiao, 2024) combines a seasonal-trend decomposition with DDPM diffusion, using a Fourier-based training objective and deep decomposition embeddings. For the conditional generation, it employs an instance-aware guidance strategy based on adaptable target metrics.

**Flow Matching (FM)**, on the other hand, directly learns a vector field representing the velocities of a continuous time transformation, interpolating between samples from an initial density (which can be, but is not limited to, Gaussian noise) and samples from the target data density, which is unknown. The forward process is presented as a linear interpolation: $\mathbf{z}_t = (1 - t)\mathbf{x} + t\epsilon$, linking the data sample $\mathbf{x}$ with a sample from the initial density $\epsilon$. An example is the flow-based model **Flow-TS**, presented in (Hu et al., 2025), in which the authors use a rectified flow with ODE-based straight lines transport for multivariate TS generation, with an adaptive sampling strategy.

**Time-VQVAE**, presented in (Lee et al., 2023), is a two-stage modeling approach similar to the one in (Chang et al., 2022) in which a VQ-VAE is used for the first stage and MaskGIT for the second (van den Oord et al., 2017). MaskGIT trains a bidirectional Transformer model in the masked modeling manner, which accelerates the sampling process significantly. First, a projection between input and a discrete latent space is learned (i.e. *tokenization*) and secondly, a prior of the discrete tokens is learned in the discrete latent space. The authors used WaveNet (van den Oord et al., 2016) as a prior model when training with TS data, and chose to use MaskGIT instead of an autoregressive model as in (van den Oord et al., 2017), resulting in an extremely short inference time. However, since they work directly with the Spectrograms instead of the raw TS, it takes a long time to train the framework.

The **Structured State Space Diffusion (SSSD)** model, adapted to ECG signals in (Alcaraz & Strodthoff, 2023), is particularly relevant because it replaces standard convolutional or attention backbones with Structured State Space (S4) layers. This enables the modeling of extremely long-range dependencies, which is crucial for long medical TS, much more efficiently than Transformers or CNNs.

## 4. Our Method: Spectrally-Partitioned Flow Matching

To address these challenges, we propose **Spectrally-Partitioned Flow Matching (SPFM)**, where instead of training a single model on raw signals, we explicitly partition the data's spectral components and assign a specialized flow model to each (see Figure 1). By isolating the high-frequency component, its associated model optimizes a loss function determined solely by high-frequency errors, eliminating the gradient dominance problem described in Section 2.2. This enables **gradient democracy**: each model receives balanced gradient signals proportional to its frequency band (see Appendix B.2.1).

Furthermore, we adopt a hierarchical conditioning scheme where the Detail model is conditioned on both the class label and the low-frequency data (real when training and generated when sampling), allowing it to learn **heteroscedasticity**: noise distributions that vary based on the signal's phase, thereby improving the synthesis of biological artifacts (Engle, 1982). To do so we employ direct conditioning, deliberately avoiding Classifier-Free Guidance (CFG), which eliminates the computational overhead of dual forward passes and prevents the spectral over-sharpening artifacts often associated with guidance extrapolation.

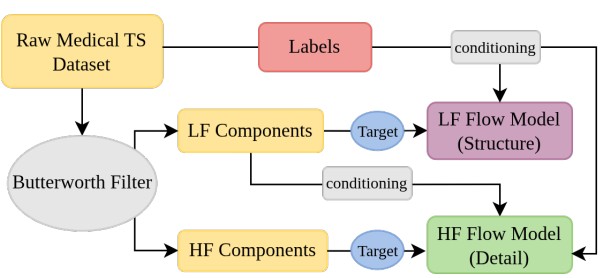

*Figure 1.* Illustration of our SPFM framework.

### 4.1. Spectral Partitioning

We decompose the raw time series data $\mathbf{x}_{\text{raw}} \in \mathbb{R}^{C \times L}$ into low-frequency structural and high-frequency detail components using a zero-phase, 4th-order Butterworth filter:

$$\mathbf{x}_{\text{LF}} := \text{ButterworthFilter}(\mathbf{x}_{\text{raw}}, f_{\text{cutoff}}) \quad (3)$$

$$\mathbf{x}_{\text{HF}} := \mathbf{x}_{\text{raw}} - \mathbf{x}_{\text{LF}} \quad (4)$$

A key design choice of this framework is the cutoff frequency $f_{\text{cutoff}}$, which was calibrated to satisfy an energy balancing criterion ensuring $\text{Var}(\mathbf{x}_{\text{LF}}) \approx \text{Var}(\mathbf{x}_{\text{HF}})$. This equilibrium is essential for preventing gradient dominance by either component during optimization. Empirically, $f_{\text{cutoff}}$ depends on the spectral decay of the signal: datasets exhibiting higher stochasticity or a shallower $1/f^{\beta}$ spectral roll-off (such as EEG) need a lower cutoff (see Section 5.1 for dataset-specific values).

### 4.2. Joint Conditional Flow Matching

SPFM employs two flow-based models parametrized by neural networks: a **Structure Model** $\phi_{\text{LF}}$ (parametrized by $\theta_{\text{LF}}$) and **Detail Model** $\phi_{\text{HF}}$ (parametrized by $\theta_{\text{HF}}$). These networks are trained to approximate the time-dependent vector fields $v_t^{\text{LF}}$ and $v_t^{\text{HF}}$ that transport the probability mass from a Gaussian noise distribution $p_0(\mathbf{z}) = \mathcal{N}(\mathbf{0}, \mathbf{I})$ to the data distribution. We adopt the Optimal Transport conditional flow matching path, whose training objective minimizes the difference between the network output and the target velocity field $\mathbf{u}_t(\mathbf{x}|\mathbf{x}_1) = \mathbf{x}_1 - \mathbf{z}$:

$$\mathcal{L}_{\text{LF}}(\theta_{\text{LF}}) = \mathbb{E}_{t,p(\mathbf{z}),q_{\text{LF}}}\left[||v_t^{\text{LF}}(\mathbf{x}_t^{\text{LF}}, y) - (\mathbf{x}_1^{\text{LF}} - \mathbf{z})||_{p_{\text{LF}}}^{p_{\text{LF}}}\right]$$

$$\mathcal{L}_{\text{HF}}(\theta_{\text{HF}}) = \mathbb{E}_{t,p(\mathbf{z}),q_{\text{HF}}}\left[||v_t^{\text{HF}}(\mathbf{x}_t^{\text{HF}}, \mathbf{x}_1^{\text{LF}}, y) - (\mathbf{x}_1^{\text{HF}} - \mathbf{z})||_{p_{\text{HF}}}^{p_{\text{HF}}}\right]$$

where $\mathbf{x}_t = (1-t)\mathbf{z} + t\mathbf{x}_1$, expectations are over $t \sim U[0,1]$, and $(p_{\text{LF}}, p_{\text{HF}})$ denote the exponents of the $\mathcal{L}_p$ loss norms. The Structure Model generates global trends conditioned on class label $y$, while the Detail Model is conditioned on both $y$ and the structural signal $\mathbf{x}_1^{\text{LF}}$.

---

**Algorithm 1** SPFM Sampling

---

**Require:** $N_{\text{LF}}, N_{\text{HF}}$ timesteps, $v_{\theta_{\text{LF}}}^{\text{LF}}, v_{\theta_{\text{HF}}}^{\text{HF}}$ models , $y$ label
   Initialize $\mathbf{z} \sim \mathcal{N}(0, \mathbf{I})$
   Set $\mathbf{x}_0^{\text{LF}} = \mathbf{z}$ and $\mathbf{x}_0^{\text{HF}} = \mathbf{z}$
   **for** $t = 0$ **to** $N_{\text{LF}} - 1$ **do**
      $\mathbf{x}_{t+1}^{\text{LF}} = v_{\theta_{\text{LF}}}^{\text{LF}}(\mathbf{x}_t^{\text{LF}}|y)$
   **end for**
   Set $\hat{\mathbf{x}}_{\text{LF}} = \mathbf{x}_{N_{\text{LF}}-1}^{\text{LF}}$
   **for** $t = 0$ **to** $N_{\text{HF}} - 1$ **do**
      $\mathbf{x}_{t+1}^{\text{HF}} = v_{\theta_{\text{HF}}}^{\text{HF}}(\mathbf{x}_t^{\text{HF}}|\hat{\mathbf{x}}_{\text{LF}}, y)$
   **end for**
   Set $\hat{\mathbf{x}}_{\text{HF}} = \mathbf{x}_{N_{\text{HF}}-1}^{\text{HF}}$
**Return:** $\hat{\mathbf{x}} = \hat{\mathbf{x}}_{\text{LF}} + \hat{\mathbf{x}}_{\text{HF}}$

---

### 4.3. Scale-Specific Loss Design

Unlike monolithic models constrained to a single loss function, our framework enables scale-specific loss tailoring with joint objective $\mathcal{L}_{\text{SPFM}} = \mathcal{L}_{\text{LF}} + \mathcal{L}_{\text{HF}}$. We utilize a generalized $\mathcal{L}_p$ norm where $p_{\text{LF}}$ and $p_{\text{HF}}$ are independently tuned from $\{1, 1.5, 2\}$ (see Section 6.3). Standard $\mathcal{L}_2$ (MSE) converges to the distribution mean, often causing blurring in high-frequency components, while $\mathcal{L}_1$ (MAE) targets the median, promoting sharper transitions but occasionally sacrificing smoothness. The fractional $\mathcal{L}_{1.5}$ loss provides a middle ground, balancing baseline stability with morphological fidelity, however the optimal combination varies by dataset and intended use.

We explored incorporating explicit Fourier-domain losses but found they over-constrained the vector field, degrading smoothness without improving fidelity (see Appendix B.4).

### 4.4. Sampling and Efficiency

Our approach prioritizes computational accessibility. Using lightweight Transformer, Mamba or hybrid backbones, we maintain a modest VRAM footprint (∼6-8GB) while avoiding the massive parameter counts of Large Time Models (Jin et al., 2023). The sampling process (Algorithm 1) is sequential: first generating the low-frequency structure, then conditionally generating high-frequency details. We utilize a simple Euler ODE solver with only 40 combined sampling steps, significantly outperforming the efficiency of standard diffusion baselines.

## 5. Datasets

We validate SPFM on four publicly available medical time-series datasets, spanning Electrocardiography (ECG) and Electroencephalography (EEG). These domains characterize the small data regime, as it is sensitive, expensive to collect, and class-imbalanced: the perfect scenario where synthetic augmentation offers high utility. The full description of

*Table 1.* Dataset attributes

| Attribute | PTB-XL | Chapman | Sleep-EDF | ISRUC |
|---|---|---|---|---|
| Data type | ECG | ECG | EEG | EEG |
| Sequence length | 1000 | 1000 | 3000 | 3000 |
| Feats. (channels) | 12 | 12 | 2 | 2 |
| Classes (labels) | 7 | 7 | 5 | 5 |
| Train size | 16433 | 35103 | 65845 | 85468 |
| Test size | 2051 | 4387 | 8693 | 11180 |
| Validation size | 2050 | 4388 | 8864 | 11136 |

*Table 2.* PTB-XL SoA Comparison

| Model | Params | Sinkhorn | SMSE | FID |
|---|---|---|---|---|
| Diffusion-TS | 16.76M | 80.75 | 1.46e-6 | 1.8153 |
| Time-VQVAE | **1.33M** | 77.36 | 1.41e-6 | 0.1886 |
| Time-Diffusion | 10.41M | 84.20 | 1.61e-6 | 0.1851 |
| SSSD-ECG | 26.43M | 78.03 | 1.29e-6 | 0.0633 |
| **SPFM Models (ours)** | | | | |
| $\mathcal{L}_1, \mathcal{L}_2$ (Mamfor.) | 3.59M | **76.85** | 1.44e-6 | 0.0590 |
| $\mathcal{L}_2, \mathcal{L}_1$ (Mamfor.) | 3.59M | 81.22 | 7.66e-7 | **0.0207** |
| $\mathcal{L}_{1.5}, \mathcal{L}_2$ (Mamba) | 3.48M | 114.12 | **1.78e-8** | 0.1614 |

the datasets, including acquisition and preprocessing details are in the Appendix C, and here we present a short table summarizing of their attributes: 1.

### 5.1. Data Processing Through Spectral Decomposition

Targets $\mathbf{x}_{LF}$ and $\mathbf{x}_{HF}$ are generated via a zero-phase, 4th-order Butterworth low-pass filter. To prevent gradient dominance, we select specific filter cutoffs that equalize the energy of the resulting low- and high-frequency targets ($\text{Var}(\mathbf{x}_{LF}) \approx \text{Var}(\mathbf{x}_{HF})$). Empirically, this equilibrium requires a cutoff of $\sim 6.25$ Hz for ECG, whereas the more stochastic Sleep EEG data requires a lower threshold of $\sim 4.30$ Hz. The corresponding power spectral densities (Appendix B) highlight this disparity: while ECG energy is concentrated in structural harmonics, EEG exhibits a broad-spectrum power distribution that complicates high-frequency reconstruction.

## 6. Experiments

We now present a comprehensive empirical validation of the SPFM framework across four medical TS datasets. We begin by defining a multidimensional assessment protocol in Section 6.1, designed to quantify performance across geometric, spectral, and semantic metrics. To establish the efficacy of our approach, we first benchmark SPFM against SoA generative baselines in Section 6.2, focusing on the trade-off between fidelity and computational efficiency. Subsequently, Section 6.3 details a granular ablation study, analyzing the impact of backbone architectures and dual-loss compositions on convergence and metric quality. Finally, we demonstrate the practical utility of our synthetic data via a downstream ECG experiment in Section 7.

### 6.1. Evaluation Metrics

We assess the quality of generated samples using three complementary metrics covering distribution, frequency, and semantic fidelity:

1. **Sinkhorn Distance:** An entropically regularized approximation of the Wasserstein Distance (Optimal Transport). It measures the geometric cost to align

the synthetic and real multivariate distributions, providing a robust gauge of structural similarity.

2. **Spectral Mean Squared Error (SMSE):** To measure frequency domain fidelity, we compute the MSE between the logarithmic Power Spectral Densities (PSD) of real and generated signals: $\text{MSE}(\log(1 + \text{PSD}_{real}), \log(1 + \text{PSD}_{syn}))$. The log-scale ensures HF components are not drowned out by high-energy LF ones.

3. **MOMENT FID:** To evaluate perceptual fidelity and diversity, we compute the Fréchet Inception Distance (FID) (Heusel et al., 2017) using the embeddings from the pretrained **MOMENT** (Goswami et al., 2024) embedding model, a SoA open-source TS foundation model. We extract the embeddings via a sliding window approach, ensuring the metric captures high-level semantic temporal features.

More details can be found in the Appendix D.1. To ensure a fair and consistent evaluation, all generated synthetic datasets perfectly mirror the class distribution of the first 100 batches of the test partition of each real dataset.

### 6.2. Comparison to other SoA ECG GM Methods

To compare the efficacy of our proposed framework, we benchmarked it against leading generative models for multivariate ECG synthesis, specifically focusing on the trade-off between generation fidelity and computational resource requirements. Our evaluation includes high-performing baselines from the diffusion paradigm, each with a different backbone architecture: SSSD-ECG (State-Space), Time-Diffusion (U-Net) [1], and Diffusion-TS (Transformer), as well as the autoregressive paradigm, represented by Time-VQVAE.

We show the metrics for three of our trained models from the ablation study 6.3, namely the best model with respect to each of the evaluation metrics in Tables 2 and 3. We can

---

[1]For Time-Diffusion, we employ the DDPM framework (Ho et al., 2020) with a 1D UNet backbone adapted for TS data.

Table 3. Chapman SoA Comparison

| Model | Params | Sinkhorn | SMSE | FID |
|---|---|---|---|---|
| Diffusion-TS | 16.76M | 85.06 | 1.56e-6 | 1.7538 |
| Time-VQVAE | **1.33M** | **77.89** | 1.34e-6 | 0.2554 |
| Time-Diffusion | 10.41M | 112.90 | 5.72e-7 | 0.0847 |
| SSSD-ECG | 26.43M | 205.68 | 4.05e-6 | 0.4156 |
| **SPFM Models (ours)** | | | | |
| $\mathcal{L}_{1.5}, \mathcal{L}_1$ (Mamfor.) | 3.59M | 196.20 | 3.98e-6 | 0.0416 |
| $\mathcal{L}_2, \mathcal{L}_1$ (Mamba) | 3.48M | 332.46 | **6.06e-8** | 0.1442 |
| $\mathcal{L}_{1.5}, \mathcal{L}_2$ (Mamfor.) | 3.59M | 201.47 | 3.29e-6 | **0.0321** |

Table 4. ECG Datasets FID Optimal Configurations

| Backbone | Step | Losses | Sinkhorn | SMSE | FID |
|---|---|---|---|---|---|
| **PTB-XL Dataset** | | | | | |
| Trans. | 50K | $\mathcal{L}_{1.5}, \mathcal{L}_{1.5}$ | 368.18 | 4.40e-06 | 0.0633 |
| Mamba | 75K | $\mathcal{L}_2, \mathcal{L}_2$ | 121.88 | **2.65e-08** | 0.1160 |
| Mamfor. | 100K | $\mathcal{L}_1, \mathcal{L}_1$ | **81.22** | 7.66e-07 | **0.0207** |
| **Chapman Dataset** | | | | | |
| Transf. | 75K | $\mathcal{L}_{1.5}, \mathcal{L}_{1.5}$ | 307.10 | 8.42e-07 | 0.0532 |
| Mamba | 100K | $\mathcal{L}_2, \mathcal{L}_1$ | 289.76 | **3.44e-07** | 0.1205 |
| Mamfor. | 50K | $\mathcal{L}_{1.5}, \mathcal{L}_1$ | **201.47** | 3.29e-06 | **0.0321** |

Table 5. ECG Datasets SMSE Optimal Configurations

| Backbone | Step | Losses | Sinkhorn | SMSE | FID |
|---|---|---|---|---|---|
| **PTB-XL Dataset** | | | | | |
| Transf. | 50K | $\mathcal{L}_1, \mathcal{L}_2$ | 125.40 | 9.54e-08 | 0.0717 |
| Mamba | 75K | $\mathcal{L}_{1.5}, \mathcal{L}_2$ | 114.12 | **1.78e-08** | 0.1614 |
| Mamfor. | 50K | $\mathcal{L}_2, \mathcal{L}_1$ | **90.61** | 3.98e-07 | **0.0314** |
| **Chapman Dataset** | | | | | |
| Transf. | 100K | $\mathcal{L}_2, \mathcal{L}_2$ | 340.54 | 1.17e-7 | **0.0835** |
| Mamba | 25K | $\mathcal{L}_2, \mathcal{L}_1$ | 332.46 | **6.06e-8** | 0.1443 |
| Mamfor. | 100K | $\mathcal{L}_2, \mathcal{L}_2$ | **322.17** | 5.01e-7 | 0.1401 |

conclude from that our framework excels the SoA methods across virtually all evaluated metrics for both ECG datasets. Specifically, our approach demonstrates superior spectral fidelity, as evidenced by lower SMSE scores as well as retaining visual realism and coherence by keeping the FID lower than the baselines. Furthermore, we can tell that our trained models require a fraction of the parameters compared to other SoA methods (excluding Time-VQVAE) and are much faster during inference, making SPFM suitable for real-time applications and deployment on devices with limited VRAM.

## 6.3. Architectural and Loss Sensitivity Analysis

Now that we have established the competitiveness of our framework, we show the results of an extensive ablation study across three distinct backbone architectures: a standard **Transformer**, a Selective State Space Model (**Mamba**) and a hybrid **Mamformer** [2]. For this we follow Algorithm 1, using the Euler ODE solver with a budget of 20 Function Evaluations (NFE) for each of the $\phi_{LF}$ and $\phi_{HF}$ trained models, resulting in a total of 40 NFEs. Given the multi-objective nature of our loss function, we analyzed the performance across all 9 combination of the loss objectives described in Section 4.3. Our analysis focuses on three primary dimensions:

- **Architectural Comparison:** How different backbones handle the tradeoff between structural integrity and high-frequency detail.

- **Training Dynamics:** Models were trained for a total of 100K steps, with checkpoints saved every 25K steps to monitor convergence.

- **Dual-Loss Selection:** Identifying the optimal $(\mathcal{L}_{LF}, \mathcal{L}_{HF})$ combination for high-fidelity medical TS synthesis, w.r.t. each of the metrics.

Tables 4 and 5 summarize the optimal configuration for the

---

[2]This architecture interleaves self-attention (Transformer) layers with bidirectional Mamba blocks (1:3 ratio).

ECG datasets with respect to the FID and SMSE metrics respectively, and tables 6 and 7 do the same for the EEG datasets. We address each of the aforementioned dimensions in sections 6.3.1, 6.3.2, and 6.3.3. Additionally, we include the optimal configuration for minimizing the Sinkhorn distance in the Appendix, Tables 9 and 10 as well as their due analysis in Section D.2.

### 6.3.1. ARCHITECTURAL COMPARISON

Across the ECG datasets, the **Mamformer** backbone consistently achieves the lowest FID scores (0.0207 on PTB-XL and 0.0321 on Chapman). This suggests that the hybrid architecture successfully combines Mamba's efficiency in modeling long-range dependencies with the Transformer's attention mechanism, which is crucial for capturing complex morphological features like the P-wave and QRS complex. While the **Mamba** backbone proved superior in terms of spectral fidelity (converging to the lowest SMSE in only 25K steps), this often came at the cost of semantic fidelity (higher FID). This indicates that while SSMs are excellent at replicating periodic frequency oscillations, they benefit significantly from the attention mechanism to spatially align these features.

In the EEG domain, the dominance of the hybrid approach was even more pronounced. Contrary to single-model baselines, the **Mamformer** achieved the best FID for Sleep-EDF (0.0796) the **Transformer** for ISRUC (0.0384), with the Mamformer slightly behind at (0.0564). Notably, the opti-

*Table 6.* EEG Datasets FID Optimal Configurations

| Backbone | Step | Losses | Sinkhorn | SMSE | FID |
|---|---|---|---|---|---|
| **Sleep-EDF Dataset** | | | | | |
| Transf. | 100K | $\mathcal{L}_2, \mathcal{L}_2$ | 3671.14 | 1.31e-3 | 0.3725 |
| Mamba | 75K | $\mathcal{L}_1, \mathcal{L}_1$ | 3594.45 | 1.08e-3 | 0.2857 |
| Mamfor. | 25K | $\mathcal{L}_{1.5}, \mathcal{L}_2$ | **3289.60** | **8.97e-4** | **0.0796** |
| **ISRUC Dataset** | | | | | |
| Transf. | 75K | $\mathcal{L}_2, \mathcal{L}_2$ | **2887.30** | 2.49e-3 | **0.0384** |
| Mamba | 75K | $\mathcal{L}_2, \mathcal{L}_1$ | 4820.28 | **3.54e-4** | 0.0702 |
| Mamfor. | 75K | $\mathcal{L}_{1.5}, \mathcal{L}_1$ | 3359.06 | 1.21e-3 | 0.0564 |

*Table 7.* EEG Datasets SMSE Optimal Configurations

| Backbone | Step | Losses | Sinkhorn | SMSE | FID |
|---|---|---|---|---|---|
| **Sleep-EDF Dataset** | | | | | |
| Transf. | 25K | $\mathcal{L}_2, \mathcal{L}_2$ | 3933.39 | 8.02e-4 | **0.5199** |
| Mamba | 75K | $\mathcal{L}_{1.5}, \mathcal{L}_{1.5}$ | 4513.60 | **2.00e-4** | 0.6623 |
| Mamfor. | 25K | $\mathcal{L}_2, \mathcal{L}_2$ | **3683.47** | 7.22e-4 | 1.0759 |
| **ISRUC Dataset** | | | | | |
| Transf. | 25K | $\mathcal{L}_2, \mathcal{L}_{1.5}$ | **3538.24** | 1.74e-3 | 0.2816 |
| Mamba | 75K | $\mathcal{L}_2, \mathcal{L}_2$ | 4643.95 | **2.96e-4** | **0.0914** |
| Mamfor. | 25K | $\mathcal{L}_2, \mathcal{L}_1$ | 4830.78 | 3.25e-4 | 0.2198 |

mal performance in these high-entropy datasets was often unlocked not just by the architecture, but by its pairing with the $\mathcal{L}_{1.5}$ loss (as detailed below), which allowed the model to balance the noise-suppression of MSE with the edge-preservation of MAE.

### 6.3.2. CONVERGENCE

Across all four datasets, a consistent lopsided convergence pattern emerges: the SMSE metric typically stabilizes earlier in the training process than the FID. In both ECG and EEG data, the Mamba and Mamformer backbones often reach near-optimal spectral fidelity by the 25K or 50K step mark. This confirms that the SPFM framework provides a solution to the Spectral Bias problem, learning the global frequency distribution almost immediately. Conversely, the FID continues to improve significantly until the 75K or 100K mark.

This lag supports our hypothesis regarding the decoupling of **Structure** and **Detail**: the model first acquires the LF baseline (minimizing spectral error), and only in the later training stages does the Detail model fully learn to phase-lock the HF transients to the structural components. The Mamformer exhibits the most stable convergence profile throughout this process, avoiding the late-stage fluctuations observed in the pure Transformer architecture on the sleep EEG data.

### 6.3.3. DUAL LOSS COMPARISON

A distinct performance tradeoff exists between distributional realism (FID) and spectral precision (SMSE), governed largely by the choice of $\mathcal{L}_p$ norm. For the **Structure (LF)** component, we observed that datasets with high morphological variance (such as Chapman and ISRUC) benefited most from the $\mathcal{L}_{1.5}$ loss. This fractional norm effectively bridges the gap between stability and sharpness, enforcing more structural integrity than $\mathcal{L}_2$ without the constraints of $\mathcal{L}_1$. For the Detail (HF) model, $\mathcal{L}_1$ consistently improved the FID by preserving stochastic textures, whereas $\mathcal{L}_2$ often led to over-smoothing, benefiting SMSE but degrading perceptual quality.

Overall, the optimal configuration is task-dependent. For applications requiring clinical realism (e.g., synthetic data augmentation), the Mamformer with a mixed configuration, typically $(\mathcal{L}_{1.5}, \mathcal{L}_1)$ or $(\mathcal{L}_2, \mathcal{L}_1)$, should be prioritized to minimize the Sinkhorn distance. Conversely, for downstream signal processing tasks such as Heart Rate Variability (HRV) or spectral analysis, a configuration like the Mamba backbone with $(\mathcal{L}_2, \mathcal{L}_1)$ losses is likely to be more effective, achieving spectral convergence with significantly fewer training steps.

## 7. Practical Utility in a Downstream Task

To demonstrate the clinical utility of the SPFM framework, we evaluated its capacity for **Spectral Super-Resolution** (bandwidth extension). Similar to audio super-resolution tasks, the goal is to generate (hallucinate) plausible HF details conditioned solely on the band-limited LF structure of the data. This is particularly relevant for remote patient monitoring, where low-cost, low-power wearable devices capture LF signals which must be upsampled to diagnostic qulaity by reconstructing the missing HF components (Kuleshov et al., 2017).

We frame this as a conditional generation problem where the Structure Model is bypassed, and the pre-trained Detail Model is used to recover the missing frequency content $\mathbf{x}_{HF}$ given a ground-truth low-frequency input $\mathbf{x}_{LF}$ from the test set, using the RK4 ODE solver with 10 steps (consistent with our constraint of 40 NFEs). We validate this on the PTB-XL and Chapman ECG datasets, measuring reconstruction fidelity using the Mean Squared Error (MSE) between the true and reconstructed HF components. As a baseline, we first compute the MSE between the real data and the null hypothesis: using only the LF components, masking out the HF components.

Figures 2 and 3 visualize the spectral super-resolution results. The top panels compare the ground truth against the super-resolved signal ($\hat{\mathbf{x}} = \mathbf{x}_{LF} + \hat{\mathbf{x}}_{HF}$) across all timesteps, while the bottom panels isolate the generated high-frequency

residuals for a reduced amount of timesteps. Quantitative results in Table 8 confirm that SPFM succeeds in recovering the majority of the missing frequencies across both Test and Validation sets. By successfully restoring the missing spectral energy, the model recovers critical diagnostic morphology, including sharp QRS complexes and low-amplitude P-waves, thus demonstrating its potential as a high-generative compression scheme for cardiac telemonitoring. Furthermore, this method also serves as a pure data augmentation technique, suitable for cases such as class imbalance or small sample size.

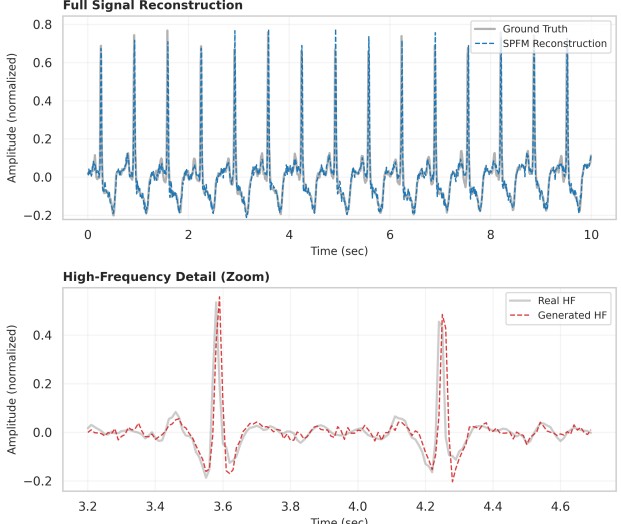

*Figure 2.* HF Reconstruction with the Detail model (Mamformer $\mathcal{L}_2$) for a PTBXL myocardial infarction (lead I) data sample.

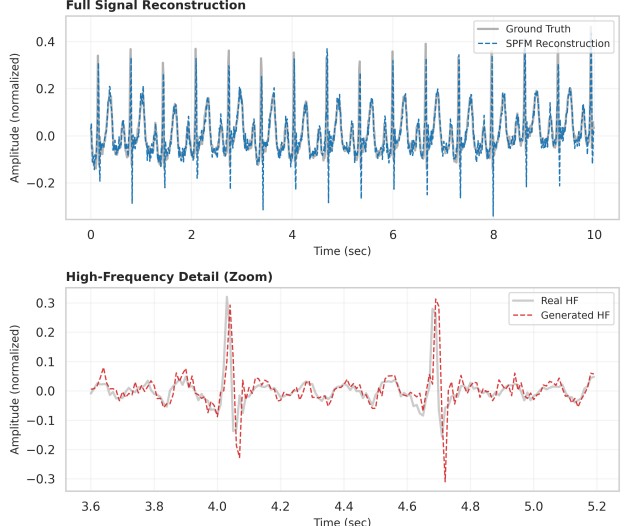

*Figure 3.* HF Reconstruction with the Detail model (Transformer $\mathcal{L}_2$) for a Chapman myocardial infarction (lead II) data sample.

*Table 8.* Quantitative evaluation of Spectral Super-Resolution for partitions of ECG datasets. The Baseline MSE represents the error of the null hypothesis (LF-only).

| Dataset | Split | Method | MSE | $\triangle$MSE |
|---|---|---|---|---|
| **PTB-XL** | *Test* | Baseline | 0.01843 | - |
| | | **SPFM** | **0.002455** | **86.68%** |
| | *Val* | Baseline | 0.01828 | - |
| | | **SPFM** | **0.002569** | **85.95%** |
| **Chapman** | *Test* | Baseline | 0.02864 | - |
| | | **SPFM** | **0.009829** | **65.68%** |
| | *Val* | Baseline | 0.02808 | - |
| | | **SPFM** | **0.008573** | **69.47%** |

## 8. Conclusion

In this work, we identified gradient dominance as a primary cause of spectral degradation in generative modeling for medical time series. We introduced Spectrally-Partitioned Flow Matching (SPFM), a lightweight yet powerful framework that enforces gradient democracy by explicitly decoupling the generation of low- and high-frequency components. Through rigorous evaluation on four diverse datasets (PTB-XL, Chapman, Sleep-EDF, and ISRUC), we demonstrated that SPFM outperforms SoA GM baselines in semantic (FID), spectral (SMSE), and geometric (Sinkhorn) fidelity while requiring significantly fewer parameters.

Our ablation studies identify the Mamformer backbone as uniquely capable of mitigating spectral shortcomings of current GM methods, and we found that a fractional $\mathcal{L}_{1.5}$ loss sometimes performs better than the usual $\mathcal{L}_1$ (MAE) and $\mathcal{L}_2$ (MSE) losses, suggesting a more thorough investigation in how general losses affect the training of the models and the quality of the generated samples. Lastly, SPFM offers immediate practical utility: the model produces not just visually coherent but also semantically robust signals, serving as a powerful engine for spectral super-resolution and data augmentation, with more use cases readily available without the need of retraining the models.

## 9. Future Work

While our current framework relies on fixed spectral partitioning, future iterations of SPFM will investigate learnable frequency decomposition, where the model dynamically optimizes the cutoff frequency based on external factors such as subject metadata. Furthermore, we plan to extend the architecture to multivariate, cross-modal biosignals (e.g. simultaneous ECG and PPG generation), aiming to model inter-channel correlations without the interference of noise.

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

# A. Theory Behind our Framework

## A.1. Conditioning and Probability Paths

In general, as we have shown in Section 2, the goal in generative modeling is to transform source samples $X_0 \sim p$ (for $p$ a simple prior distribution, e.g. Gaussian noise) to target samples $X_1 \sim q$, where $q$ is the target distribution. In standard (unconditional) diffusion models, the training pairs $(X_0, X_1)$ are sampled independently from the source and target distribution, i.e. $(X_0, X_1) \sim p(X_0)q(X_1)$. This transformation can be viewed as a continuous evolution of probability distributions varying over time $p_t$, for time $t \in [0, 1]$, often called a **probability path** (Lipman et al., 2024). This path connects the prior and target distributions $p_0 = p$ and $p_1 = q$, respectively. In our scenario, the *full data distribution* $q$ is described by the joint probability distribution $p(\mathbf{x}_1^{\mathrm{LF}}, \mathbf{x}_1^{\mathrm{HF}})$, which captures both the low- and high-frequency marginal distributions of the data: $p_{\mathrm{LF}}$ and $p_{\mathrm{HF}}$.

It is quite intuitive to try to learn a single probability path from a Gaussian noise distribution: $p_0 \sim \mathcal{N}(0, \mathbf{I})$ to the (target) data distribution $p_{\mathrm{data}}$ which is *squeezed* through the low-frequency distribution $p_{\mathrm{LF}}$ at some intermediate $t_{\mathrm{LF}} \in (0, 1)$. This would constrain the path $p_t$ such that $p_{t_{\mathrm{LF}}} \approx p_{\mathrm{LF}}$, forcing the model to learn the low-frequency structure on its way to generating the full, high-fidelity samples. However, when put into practice we noticed that this approach presents theoretical and practical challenges, motivating our new, conditional and spectrally-partitioned two-model framework.

This new approach redefines the probability path, where we now define two paths matching the same noise vector $\mathbf{z}$ to both the low- and high-frequency components of the samples, resulting in two learned vector fields $v_t$ that are a function of both the current state $\mathbf{x}_t$ and the conditioning components, defined by their respective marginal paths:

$$\text{Structural (LF) marginal path} := \frac{d\mathbf{x}_t}{dt} = \mathbf{v}_t^{\mathrm{LF}}(\mathbf{x}_t, y) \tag{5}$$

$$\text{Detail (HF) marginal path} := \frac{d\mathbf{x}_t}{dt} = \mathbf{v}_t^{\mathrm{HF}}(\mathbf{x}_t, \mathbf{x}_1^{\mathrm{LF}}, y) \tag{6}$$

To summarize, we obtain paired data samples $(\mathbf{x}_1^{\mathrm{LF}}, \mathbf{x}_1^{\mathrm{HF}})$ as described in Section 5.1. Then, we train two distinct CFM models, both originating from the same initial noise sample $\mathbf{z}$ and each with their own objective, anchoring the generative process of both probability paths. This ensures that the generated $\mathbf{x}_{\mathrm{LF}}$ from the structure model and the $\mathbf{x}_{\mathrm{HF}}$ from the detail model are semantically consistent and coupled. This factorization of the generation process was inspired by work on conditional flow matching (Lipman et al., 2023) and stochastic interpolants (Albergo et al., 2023).

## A.2. Theoretical Formulation of our Method

We use the modified FM objective described in 4.3 which learns a velocity field $\mathbf{v}_t^{\mathrm{LF}}$ that transports samples from a simple prior distribution (Gaussian noise $\mathbf{z} \sim \mathcal{N}(0, \mathbf{I})$ to the target LF distribution $p_{\mathrm{data}}(\mathbf{x}_{\mathrm{LF}}|y)$ conditionally. The loss is an $\mathcal{L}_p$ error between the predicted velocity $v_t^{\mathrm{LF}}(\mathbf{x})$ and the target velocity $u_t(\mathbf{x}|\mathbf{x}_1^{\mathrm{LF}}, y)$ which induces a probability path $p_t^{\mathrm{LF}}$ via the interpolation path $\mathbf{x}_t = (1 - t)\mathbf{z} + t\mathbf{x}_t^{\mathrm{LF}}$ given parameters $\theta_{\mathrm{LF}}$:

$$\mathcal{L}_{\mathrm{LF}}(\theta_{\mathrm{LF}}) = \mathbb{E}_{t, q_{\mathrm{LF}}(\mathbf{x}_1^{\mathrm{LF}})p_t^{\mathrm{LF}}(\mathbf{x}|\mathbf{x}_1^{\mathrm{LF}}, y)} \left[ ||v_t^{\mathrm{LF}}(\mathbf{x}, y) - u_t(\mathbf{x}|\mathbf{x}_1^{\mathrm{LF}}, y)||_p^p \right], \tag{7}$$

where $t$ represents the timesteps of each of the learned interpolation paths, sampled uniformly from the interval $[0, 1]$, and $p_1^{\mathrm{LF}}(\mathbf{x}) \approx q_{\mathrm{LF}}(\mathbf{x})$.

On the other hand, the Detail model, which learns the distribution $p(\mathbf{x}_{\mathrm{HF}}|\mathbf{x}_{\mathrm{LF}}, y)$, generates only the high-frequency components of the raw data, $\mathbf{x}_{\mathrm{HF}}$, conditioned on both the corresponding LF signal $\mathbf{x}_{\mathrm{LF}}$ and the class labels $y$. Like its Structure counterpart, it also uses an $\mathcal{L}_p$ loss, learning a separate velocity field $v_t^{\mathrm{HF}}$ that transports noise $\mathbf{z}$ to the target HF distribution, in the probability path $p_t^{\mathrm{HF}}$ induced by the HF target velocity field $u_t(\mathbf{x}|\mathbf{x}_1^{\mathrm{LF}}, \mathbf{x}_{\mathrm{HF}}, y)$ via the interpolation path $\mathbf{x}_t = (1 - t)\mathbf{z} + t\mathbf{x}_1^{\mathrm{HF}}$ and parameters $\theta_{\mathrm{HF}}$:

$$\mathcal{L}_{\mathrm{HF}}(\theta_{\mathrm{HF}}) = \mathbb{E}_{t, q_{\mathrm{HF}}(\mathbf{x}_1^{\mathrm{HF}}), p_t^{\mathrm{HF}}(\mathbf{x}|\mathbf{x}_1^{\mathrm{HF}}, y)} \left[ ||v_t^{\mathrm{HF}}(\mathbf{x}, \mathbf{x}_{\mathrm{LF}}, y) - u_t(\mathbf{x}|\mathbf{x}_1^{\mathrm{HF}}, \mathbf{x}_1^{\mathrm{LF}}, y)||_p^p \right], \tag{8}$$

where again, $t$ is defined as in 7 and $p_1^{\mathrm{HF}} \approx q_{\mathrm{HF}}(\mathbf{x})$.

These formulation is theoretically sound but impractical, as we do not know any of the following: $p_t^{\mathrm{LF}}, p_t^{\mathrm{HF}}, u_t^{\mathrm{LF}}, u_t^{\mathrm{HF}}$. The Flow Matching formulation, first presented in (Lipman et al., 2023) and explained thoroughly in (Lipman et al., 2024),

provides us with a key insight. Basically, we can bypass this impediment by defining a simple, known path. The key insight is how we can use the (linear) interpolation paths, $\mathbf{x}_t = (1-t)\mathbf{z} + t\mathbf{x}_{\text{LF}}$ and $\mathbf{x}_t = (1-t)\mathbf{z} + t\mathbf{x}_{\text{HF}}$, to simplify the objective. These straight paths deterministically connect a single sample from our source distribution $\mathbf{z} \sim p_0$ to a pair of our target data samples $\mathbf{x}_1^{\text{LF}}$ and $\mathbf{x}_1^{\text{HF}}$. Since we have defined these path, we can calculate their exact velocity analytically, which are their first order time derivatives:

$$u_t^{\text{LF}} = \frac{d\mathbf{x}_t}{dt} = \mathbf{x}_1^{\text{LF}} - \mathbf{z} \tag{9}$$

$$u_t^{\text{HF}} = \frac{d\mathbf{x}_t}{dt} = \mathbf{x}_1^{\text{HF}} - \mathbf{z} \tag{10}$$

This simplifies the objective by allowing us to use a constant vector as the objective, turning the intractable expectations over $p_t^{\text{LF}}(\mathbf{x})$ and $p_t^{\text{HF}}(\mathbf{x})$ into simple, practical expectations over the pairs of endpoints $(\mathbf{z}, \mathbf{x}_1^{\text{LF}})$, $(\mathbf{z}, \mathbf{x}_1^{\text{LF}})$ and time $t$.

Finally, we arrive at our simple, practical loss formulations, where we reintroduce the class labels $y$ which condition both models:

$$\mathcal{L}_{\text{LF}}(\theta_{\text{LF}}) = \mathbb{E}_{t,p(\mathbf{z}),q_{\text{LF}}(\mathbf{x}_1^{\text{LF}},y)}\left[||v_t^{\text{LF}}(\mathbf{x}_t^{\text{LF}}, y) - (\mathbf{x}_1^{\text{LF}} - \mathbf{z})||_p^p\right]$$

$$\mathcal{L}_{\text{HF}}(\theta_{\text{HF}}) = \mathbb{E}_{t,p(\mathbf{z}),q_{\text{HF}}(\mathbf{x}_1^{\text{HF}},\mathbf{x}_1^{\text{LF}},y)}\left[||v_t^{\text{HF}}(\mathbf{x}_t^{\text{HF}}, \mathbf{x}_1^{\text{LF}}, y) - (\mathbf{x}_1^{\text{HF}} - \mathbf{z})||_p^p\right]$$

where $\mathbf{x}_t^{\text{LF}} = (1-t)\mathbf{z} + t\mathbf{x}_1^{\text{LF}}$ and $\mathbf{x}_t^{\text{HF}} = (1-t)\mathbf{z} + t\mathbf{x}_1^{\text{HF}}$, the expectations are taken over timesteps $t \sim U[0,1]$, $p(\mathbf{z})$ represents the noise distribution, and the empirical data distributions are represented by $q_{\text{LF}}$ and $q_{\text{HF}}$.

### A.3. Model Design Choices

The choice of using two Flow Matching models for our framework is a deliberate design, exploiting the inherent strength of this architecture for our spectrally-partitioned framework. Flow models are governed by first-order PDEs stemming from optimal transport and are deterministic processes (Lipman et al., 2024). As described in the Generator Matching paper, the **generator** of such a process is a first-order operator that shifts probability mass along defined trajectories (Holderrieth et al., 2024). Given that no stochastic smoothing is imposed, the mapping between the source (noise) and target (Data) distributions is **bijective** and well-defined.

These properties were the key motivation for our dual-flow design, as for the structural path, using a flow model ensures the learned generative path is stable, efficient, and invertible. It provides a robust base sample $\mathbf{x}_{\text{LF}}$ that is a direct and deterministic function of the latent noise $\mathbf{z}$. Since the detail path starts from the same latent noise $\mathbf{z}$ and is conditioned on the output of the first path, its task reduces to finding the exact high-frequency components that correspond to this specific $(\mathbf{z}, \mathbf{x}_{\text{LF}})$ pair, ensuring an end-to-end deterministic and bijective trajectory. Unlike diffusion models, which introduce stochasticity during the sampling process itself, our dual flow approach isolates all randomness to the initial sampling of the latent (noise) vector $\mathbf{z}$.

## B. Spectral Partitioning Detailed Description

### B.1. Motivation

As mentioned before, a crucial part of our framework is the separation of the frequency spectrum into low- and high-frequencies, determined by the parameter $f_{\text{cutoff}}$ and resulting in the decomposition of the raw data: $\mathbf{x}_{\text{raw}} = \mathbf{x}_{\text{LF}} + \mathbf{x}_{\text{HF}}$, such that $\mathbf{x}_{\text{LF}}$ contains the frequencies (in Hz) in the clopen set $[0, f_{\text{cutoff}})$ and $\mathbf{x}_{\text{LF}}$ the ones in the closed set $[f_{\text{cutoff}}, 50]$. If $f_{\text{cutoff}}$ is set too low (e.g., $< 1.5$ Hz), the $\mathbf{x}_{\text{LF}}$ signal contains only the slowest oscillatory trends. Consequently, the conditioning signal lacks the morphological cues required to guide the generation of faster transients. Specifically, in physiological signals, high-frequency events, such as sleep spindles in EEG or late potentials in ECG, are often phase-locked to broader morphological structures like Delta waves or the QRS complex. If these structural anchors are excluded from the conditioning signal (by using a low cutoff), the **Detail** model loses the context necessary to correctly localize these transient events. This forces the model to memorize textures (in the form of jitters) rather than learning the conditional relationship, leading to a higher training loss and degraded fidelity during sampling.

Furthermore, the spectral composition of physiological signals is governed by a power-law decay, typically adhering to $P(f) \propto 1/f^\beta$ (where $\beta \approx 1 - 2$). This fractal, or so-called *pink noise* characteristic reflects the scale-free dynamics of

biological systems, resulting in a Power Spectral Density (PSD) where signal energy is heavily concentrated in the lower frequency bands. In the context of ECG and EEG, this implies that the vast majority of signal variance resides in the lowest spectral octaves, while high-frequency bands contain exponentially less energy but encode critical textural details, as evidenced by Figures 4 through 7.

### B.2. Technique

Therefore, we employ an **Energy Balancing** strategy to select an $f_{\text{cutoff}}$ that maximizes the informational value of the conditioning signal while maintaining a balanced variance between components. We define the optimal split as the frequency that equilibrates the signal variance, $\text{Var}(\mathbf{x}_{\text{LF}}) \approx \text{Var}(\mathbf{x}_{\text{HF}})$. This ensures that the LF model captures the full morphological *skeleton* of the signal, allowing the HF model to focus exclusively on refining the spectral texture given this rich structural context. Empirically, this optimum is found at $\approx 4.3$ Hz for sleep EEG and $\approx 6.25$ Hz for ECG.

To validate our choice of cutoff frequency, we visualize the signal energy distribution in Figures 4-7. These figures present two complementary views: the logarithmic-scaled Power Spectral Density (PSD) and the normalized Cumulative Energy Sum. For the PSD plots, we applied Bartlett's method by averaging the spectra computed across individual channels. Conversely, for the cumulative energy plots, we concatenated all signals into a single array to compute a global PSD. This cumulative curve acts as a quantitative proxy for signal variance, allowing us to pinpoint the exact frequency where the total energy is partitioned. By overlaying the chosen $f_{\text{cutoff}}$ on these distributions, we empirically demonstrate that our selection achieves a variance equilibrium—ensuring the conditioning signal $\mathbf{x}^{\text{LF}}$ retains sufficient structural power while leaving adequate textural energy for the $\mathbf{x}^{\text{HF}}$ component.

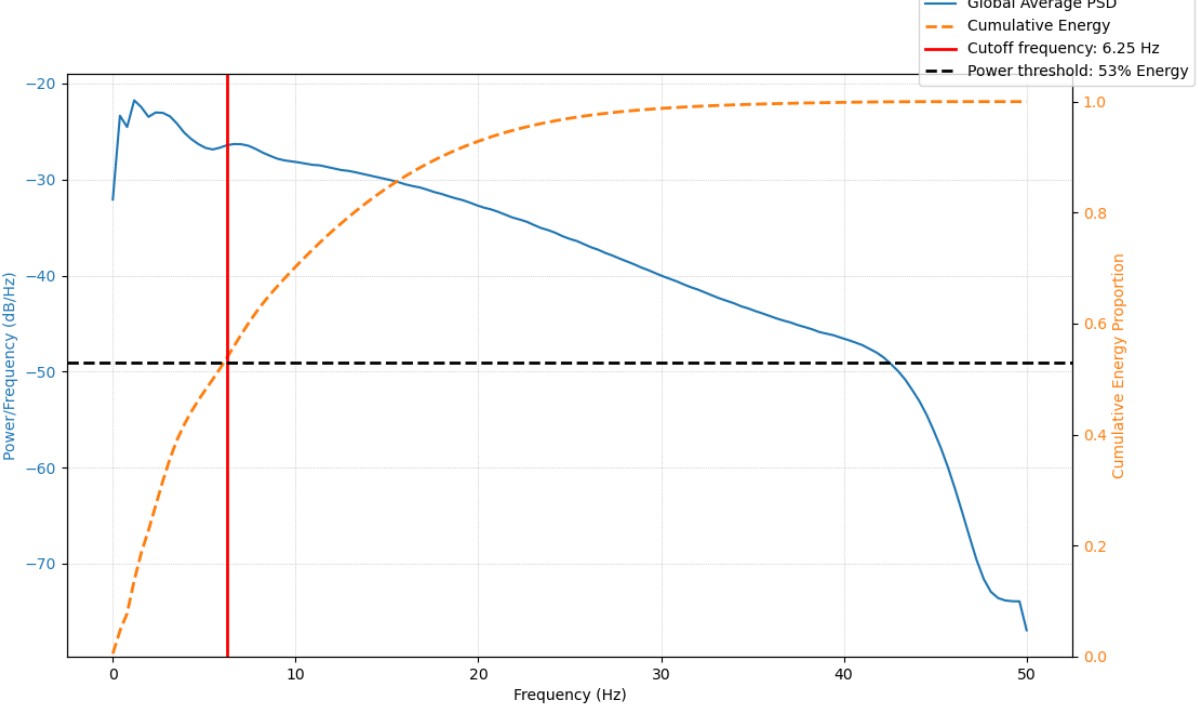

*Figure 4.* Spectral analysis and cutoff frequency selection for the PTB-XL (ECG) Dataset

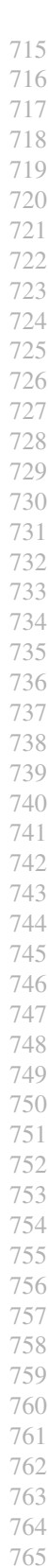

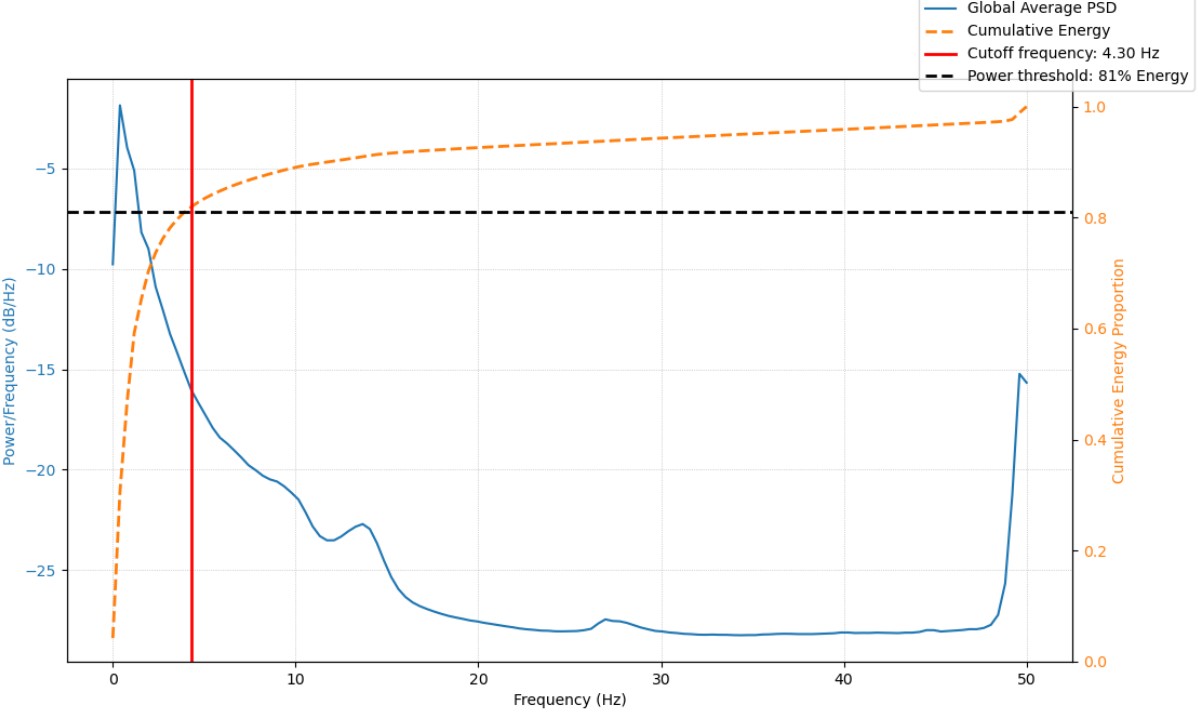

*Figure 5.* Spectral analysis and cutoff frequency selection for the Chapman (ECG) Dataset

*Figure 6.* Spectral analysis and cutoff frequency selection for the Sleep-EDF (EEG) Dataset

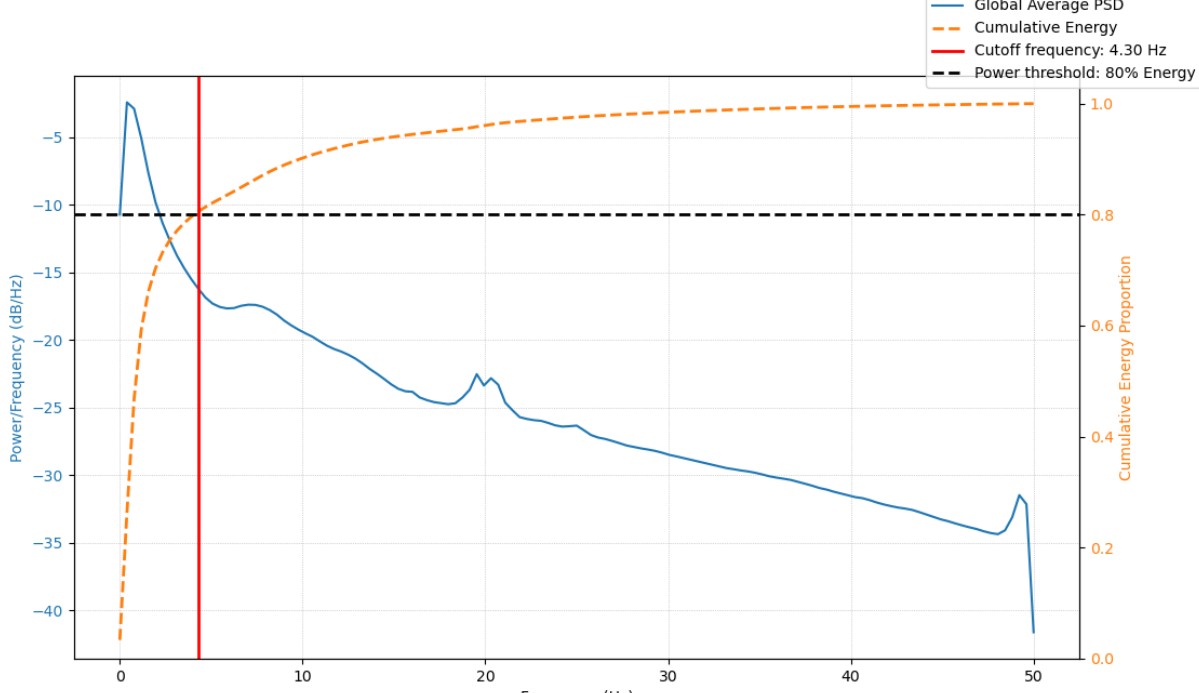

*Figure 7.* Spectral analysis and cutoff frequency selection for the ISRUC (EEG) Dataset

### B.2.1. GRADIENT DEMOCRACY: NORMALIZING THE OPTIMIZATION LANDSCAPE

The primary benefit of Energy Balancing is the emergence of what we term **gradient democracy**. In standard GM frameworks, the training objective is computed over the raw signal. Given the energy distribution inherent in TS (see Section B), the loss is disproportionally sensitive to errors in high-amplitude, low-frequency components. Mathematically, since signal variance is concentrated in the LF band, the gradient magnitude $||\nabla_\theta \mathcal{L}||$ is dominated by structural offsets. This creates a *gradient starvation* effect, where the optimizer prioritizes the global baseline and slow-moving waves because they provide the steepest path for loss reduction, effectively treating the HF details as negligible noise or residuals.

By partitioning the spectrum and training $\phi_{LF}$ and $\phi_{HF}$ on energy-equilibrated targets, we decouple these optimization scales. In the SPFM framework, the loss for the Detail model, $\mathcal{L}_{HF}$, is calculated against a signal $\mathbf{x}_{HF}$ that has been stripped of the dominant LF energy. This **democratizes** the gradients: the Detail model is forced to attend to the morphological nuances and stochastic textures of the HF band because they are no longer overshadowed by the high-amplitude structural signals. This separation ensures that the neural network's capacity is distributed across the entire spectrum, preventing the spectral bias typically observed in deep sequence models and allowing for the high-fidelity synthesis of biological artifacts (e.g., QRS notches or EEG micro-states) that would otherwise be smoothed out in other training regimes. We show a toy example of this phenomenon in Figure 8.

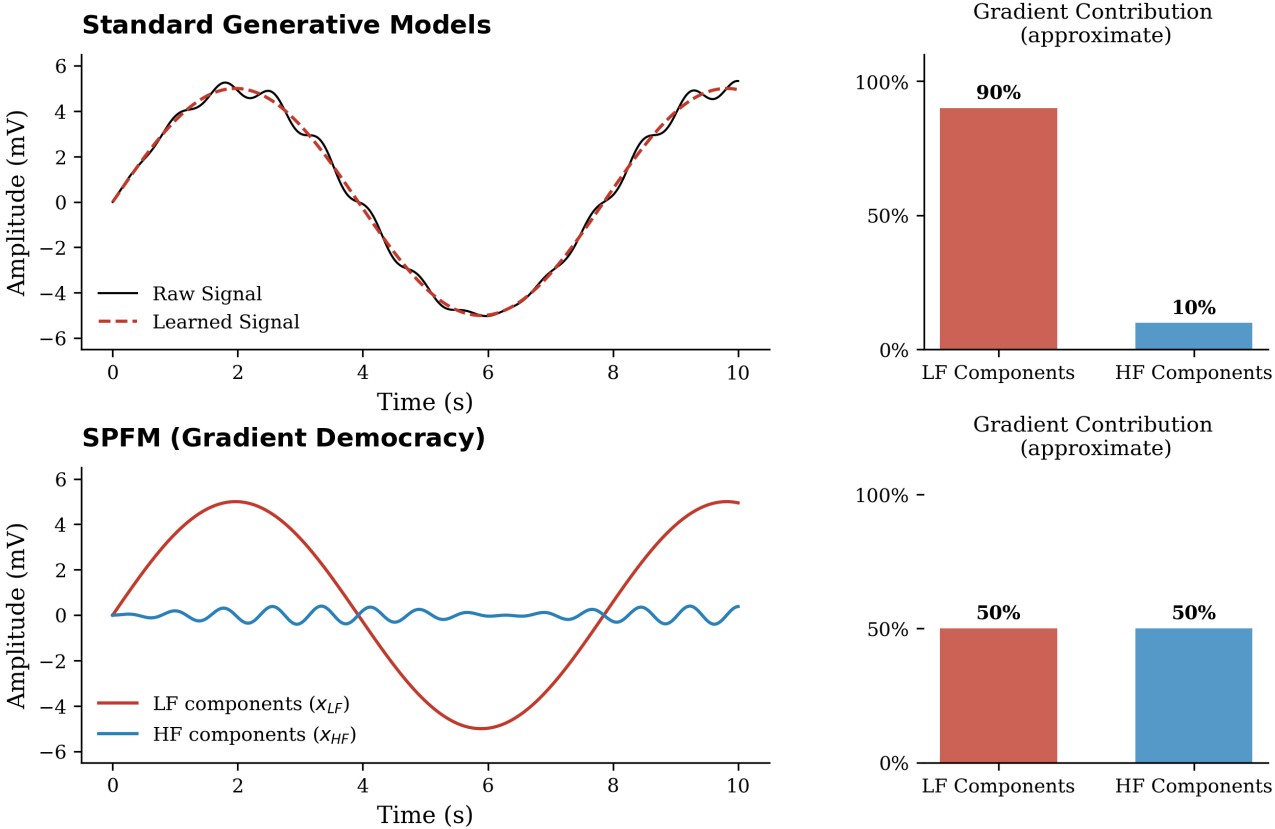

*Figure 8.* Gradient democracy visualization with toy data. We can see that the reconstructed signal of the traditional GM models is oversmoothed, due to small gradient signal from the HF components. Our SPFM formulation proposes a fix, where the gradient signal from all frequency components is treated equally, and we dedicate a specialized model for each spectral partition.

### B.3. Loss Formulation

Here we introduce the formal definition of each of the losses tested in our framework, as well as the results of the ablation study for each dataset. First, let's define the losses. The original FM objective presented in (Lipman et al., 2023) uses the $\mathcal{L}_2$ loss, which is also called the Mean-Squared Error (MSE) loss:

$$\mathcal{L}_p(\theta) = \mathbb{E}_{t,p_t(\mathbf{x})}\left[||\mathbf{v}_t(\mathbf{x}) - u_t(\mathbf{x})||_2^2\right] \tag{11}$$

We modify the original FM objective, for a target probability density path $p_t(\mathbf{x})$ and a corresponding vector field $u_t(\mathbf{x})$ (which generates $p_t$), and parameters $\theta$:

$$\mathcal{L}_p(\theta) = \mathbb{E}_{t,p_t(\mathbf{x})}\left[||\mathbf{v}_t(\mathbf{x}) - u_t(\mathbf{x})||_p^p\right], \tag{12}$$

where the expectation $\mathbb{E}[\cdot]$ is taken over time $t$ (sampled uniformly from $U[0,1]$), and we omit the conditioning factor for simplicity. For the loss ablation study 6.3, we experimented with the different $p$-values: $(1, 1.5, 2)$, experimenting with all 9 loss-combinations for our dual model framework for each of the backbone architectures studied. To the best of our knowledge, the exploration of general $\mathcal{L}_p$-losses for training FM and diffusion models is a gap in the current research landscape, as most existing frameworks settle for the well-known $\mathcal{L}_1$ or $\mathcal{L}_2$ losses. In our ablation loss experiments 6.3, we notice a huge difference in the quality of the models trained with different combinations of losses, evidencing the need for a more thorough research in how these more general losses affect the training of the models and the quality of the generated samples.

### B.4. Auxiliary Spectral Loss

While time-domain losses (e.g. MSE, MAE) effectively capture the global structure of the signals, they sometimes fail to model the complete frequency spectrum of the data, a phenomenon called the **Spectral gap** caused in part by the training dynamics of GMs, which are affected by the so-called Spectral Bias 2.2. In our experiments, we noticed that the models trained solely with time-domain losses failed to capture sharp spectral artifacts, such as the 50 Hz power-line notch filter inherent of ECG data. While these spectral features are perceptually and clinically irrelevant in the medical domain and contribute minimally to the total signal power, we still tried to see if our models could learn better with an auxiliary spectral loss. To address this, we tried incorporating an auxiliary **log-power spectral density loss ($\mathcal{L}_{\mathbf{freq}}$)** to both the Detail and Structure models. By operating in the logarithmic domain, this loss allows us to compress the dynamic range of the spectrum, equally penalizing errors in high-power (low-frequency) and low-power (high-frequency) regions.

In theory, this loss component encourages the model to faithfully reproduce the entire spectral profile, including sharp notches and low-amplitude peaks that are often ignored by standard spectral- and time-domain objectives. We define this loss as follows:

$$\mathcal{L}_{\text{freq}} = \lambda||\log(\text{PSD}(\mathbf{x}_{\text{real}})) - \log(\text{PSD}(\mathbf{x}_{\text{gen}}))||_2^2 \tag{13}$$

where $\mathbf{x}_{\text{real}}$ and $\mathbf{x}_{\text{gen}}$ are the target and interpolated (generated) samples for each model, respectively, and $\lambda$ is a scaling factor in order to balance this loss w.r.t. the time-domain one.

In summary, we tried the following experiments:

- Optimizing the joint spectral- and time-domain loss during training

- Training with the time-domain loss and finetuning with the joint loss

- Using different log SPD metrics (including a stable log1p metric)

And saw the spectral gap worsen in all of the above cases. We analyze the results by stating the potential reason to this counterintuitive but powerful result. When we generate samples, we use an ODE solver (e.g. Euler) with a fixed number of steps, which acts like a low-pass filter (to generate high-frequency oscillations the solver needs to take step sizes smaller than the wavelength of that frequency). Thus, no matter how much we punish the neural network's weights with the spectral loss component, the ODE solver simply cannot produce frequencies higher than the step size allows. Furthermore, medical sensors have thermal and quantization noise and powerline interference at high frequencies (near 50 Hz), so the real spectrum has a high *noise floor* and never drops to zero power, but ODEs (and ODEs parametrized with a neural network) are naturally smooth functions which have a much lower noise floor. Thus the spectral loss is forcing the model to learn **white noise** to match the real data's noise floor, however white noise is hard to compress and model with a flow or diffusion process.

### B.5. Insight on the Implicit Spectral Success of Time-Domain Losses

So then why are we learning the best spectral reconstruction of the signals using only a time-domain objective ($\mathcal{L}_1, \mathcal{L}_{1.5}, \mathcal{L}_2$)? We can conclude that for the datasets studied, an implicit learning beats explicit forcing. Furthermore, Parseval's theorem tells us that the energy in the time-domain is equal to the energy in the frequency domain. By minimizing the $L_p$ error between the signals: $||\mathbf{x} - \hat{\mathbf{x}}||_p^p$ we are mathematically guaranteed to minimize the $L_p$ error between their spectra: $||X(f) - \hat{X}(f)||_p^p$, meaning that when training only with the time-domain loss, we are learning the most natural, numerically stable way possible for the FM framework.

## C. Dataset Details

### C.1. ECG Datasets

Electrocardiography (ECG) serves as the primary non-invasive diagnostic tool for assessing cardiac physiology. By recording electrical activity, ECGs enable the detection of arrhythmias, a heterogeneous family of cardiac conditions characterized by irregularities in heart rate or rhythm. While expert review by cardiologists remains the gold standard, the complexity and volume of ECG data have necessitated the development of automated diagnostic systems.

The Physikalisch-Technische Bundesanstalt XL Dataset, also refered to as **PTB-XL**, introduced by (Wagner et al., 2020), was designed to advance machine learning in healthcare. It comprises 21,837 clinical 12-lead ECG recordings from 18,885 patients. Each recording is 10 seconds in duration, sampled at 500 Hz (and is available also in 100 Hz), and adheres to the SCP-ECG standard. The dataset features a rich hierarchical annotation scheme with 71 distinct classes. For the purpose of standard benchmarking, these classes are aggregated into five diagnostic superclasses: Normal (NORM), Myocardial Infarction (MI), ST/T Change (STTC), Conduction Disturbance (CD), and Hypertrophy (HYP). For all of our experiments, we use the first 8 folds of the dataset as the training data, the 9th as the test and the 10th as the validation, as in (Strodthoff et al., 2020).

The 12-lead Chapman University and Shaoxing People's Hospital, or **Chapman** for short, introduced by (Zheng et al., 2020), is another valuable resource for the analysis of cardiac rhythms. This repository contains 45,152 12-lead ECG recordings from distinct patients, maintaining the standard 10-second duration and 500 Hz sampling rate (which we have downsampled to 100 Hz). Unlike PTB-XL, the Chapman dataset focuses specifically on rhythm abnormalities. A critical advantage of this dataset is the strict one-to-one mapping between patients and recordings, which eliminates the risk of patient leakage during random partitioning, which can be a common pitfall in ECG analysis.

### C.2. EEG Datasets

Polysomnography (PSG) is the gold-standard multiparameter diagnostic tool for sleep medicine, recording simultaneous physiological signals including electroencephalogram (EEG), electrocardiogram (ECG), electrooculogram (EOG), and electromyogram (EMG). These recordings are segmented into 30-second epochs and classified into five sleep stages—Wakefulness (W), Rapid Eye Movement (REM), and Non-REM stages N1, N2, and N3—following the American Academy of Sleep Medicine (AASM) guidelines (Berry et al., 2012).

The **Sleep-EDF** database (Goldberger et al., 2000) serves as the primary benchmark for automatic sleep stage classification. We utilize the *Sleep-Cassette* subset published in 2013, which contains 39 overnight PSG recordings from 20 healthy Caucasian subjects (aged 25–101 years) obtained during a 1987-1991 study on the effects of age on sleep. Each subject was recorded for approximately 20 hours over two subsequent day-night periods at their homes. The recordings include two EEG channels, Fpz-Cz and Pz-Oz, sampled at 100 Hz. The sleep stages were originally scored according to the Rechtschaffen and Kales (R&K) standard and were subsequently mapped to the AASM standard (combining stages 3 and 4 into N3) for compatibility with modern benchmarks.

To assess model robustness on more complex clinical data, we employ the **ISRUC-Sleep** dataset, collected by the Sleep Medicine Centre of the Hospital of Coimbra University (2009-2013). This dataset consists of all-night PSG recordings from 100 adult subjects. While the original data contains 19 channels sampled at 200 Hz, we focus on the F3 and F4 EEG channels to evaluate the model's ability to learn from spatially distinct frontal derivations. We adhere to the preprocessing pipeline established by (Jia et al., 2021): signals are downsampled to 100 Hz to match Sleep-EDF, filtered with a 50 Hz notch filter to remove powerline interference, and bandpass filtered (0.3–50 Hz) using a Hamming window to isolate relevant frequency bands.

**Preprocessing**: For both EEG datasets, we implement a rigorous cleaning and normalization protocol. Movement artifacts, unknown sleep segments, and the final 30 segments of each recording are excluded. We further exclude subjects with missing channels, consistent with literature standards. Finally, we apply *subject-wise $z$-score* normalization to each EEG channel independently. This step is critical for mitigating inter-subject variability arising from acquisition hardware and anatomical differences, ensuring the model focuses on relative amplitude dynamics, essential for distinguishing high-voltage synchronous states (e.g., N3) from low-voltage desynchronous states (e.g., REM), rather than absolute signal magnitudes.

## D. SPFM Experimental Results

### D.1. SPFM Dual-Loss Evalution Metrics

Here we present a detailed description of the metrics used for the SPFM dual-loss ablation study, followed by the results for each dataset.

We quantify the statistical quality of the generated signals using a suite of established metrics:

1. **Sinkhorn Distance:** To assess multivariate distributional alignment, we compute the Sinkhorn Divergence, which is an entropically regularized approximation of the Wasserstein Distance (Optimal Transport). Unlike point-wise metrics, this measures the geometric cost required to transport the probability mass from the synthetic distribution to the real distribution. This provides a robust metric for structural similarity that is particularly effective for high-dimensional time-series data where standard Euclidean-based metrics may struggle. A lower value indicates that the distributions of the real and synthetic data are more similar.

2. **Spectral Mean Squared Error:** To evaluate the fidelity of the generated signals in the frequency domain, we employ the Spectral Mean Squared Error (SMSE). This metric quantifies the discrepancy between the global Power Spectral Densities (PSD) of the real and synthetic datasets. We estimate the PSD using Welch's method, averaged across all samples and channels to obtain a robust global spectral profile. Crucially, we apply a logarithmic transformation ($\log(1 + S)$) before calculating the Mean Squared Error, ensuring that differences in lower-power high-frequency components contribute meaningfully to the metric, preventing it from being dominated solely by high-energy low-frequency rhythms.

3. **Fréchet Inception Distance (FID):** To assess the perceptual fidelity and diversity of the generated samples, we compute a domain-adapted FID score. Since standard Inception networks are optimized for 2D natural images, we use an open-source, large model pretrained on a large collection of TS data called **MOMENT** (Goswami et al., 2024) as a domain-specific feature extractor. Since this model has a fixed context window of 512 timesteps, we use a *sliding window* approach to chop down the signals into 512-length overlapping segments. We then pass the multivariate signals to the model and extract the embeddings from the model's encoder, averaging the patch embeddings to get a single vector per window. Once we have these embeddings, we can finally compute the Fréchet distance (Heusel et al., 2017) between the real and synthetic embeddings. A lower FID indicates that the synthetic data is statistically indistinguishable from the real data in terms of high-level semantic features, effectively capturing both sample quality and mode coverage.

### D.2. Optimal Configuration for Minimizing Sinkhorn Distance

For both ECG datasets studied, the hybrid **Mamformer** architecture consistently proved superior in minimizing geometric transport costs, achieving the lowest Sinkhorn distances on both PTB-XL (76.85) and Chapman (196.20). This suggests that the Mamformer's combination of local attention and state-space modeling is particularly effective at capturing the precise morphological geometry of cardiac signals. In the EEG domain, results were more varied: while the **Mamformer** remained most effective for Sleep-EDF, the Transformer backbone achieved the lowest Sinkhorn distance on ISRUC (2705.60), significantly outperforming the other architectures.

Across all modalities, a distinct pattern emerges regarding the objective function. The optimal configuration for minimizing the Sinkhorn metric almost exclusively favors $\mathcal{L}_1$ loss (often for both Structure and Detail components), as seen in nearly every optimal entry in Tables 10 and 9. This aligns with the theoretical intuition that the $\mathcal{L}_1$-loss encourages sharper probability mass alignment and reduced blurring compared to $\mathcal{L}_2$, a property that the Optimal Transport-based Sinkhorn distance directly rewards. Finally, regarding convergence, the optimal number of steps varies significantly (ranging from 25K to 100K) and often differs from the FID- and SMSE-optimal checkpoints. This suggests that geometric convergence

*Table 9.* ECG Datasets Sinkhorn Optimal Configurations

| Backbone | Step | Losses | Sinkhorn | SMSE | FID |
|---|---|---|---|---|---|
| **PTB-XL Dataset** | | | | | |
| Transformer | 75K | $\mathcal{L}_1, \mathcal{L}_{1.5}$ | 115.75 | **1.71e-7** | 0.0734 |
| Mamba | 25K | $\mathcal{L}_1, \mathcal{L}_2$ | 83.32 | 3.71e-7 | 0.1655 |
| Mamformer | 75K | $\mathcal{L}_1, \mathcal{L}_2$ | **76.85** | 1.44e-6 | **0.0590** |
| **Chapman Dataset** | | | | | |
| Transformer | 75K | $\mathcal{L}_1, \mathcal{L}_1$ | 227.13 | 2.14e-6 | 0.0607 |
| Mamba | 25K | $\mathcal{L}_1, \mathcal{L}_2$ | 218.06 | **1.39e-6** | 0.1359 |
| Mamformer | 75K | $\mathcal{L}_{1.5}, \mathcal{L}_1$ | **196.20** | 3.78e-6 | **0.0416** |

*Table 10.* EEG Datasets Sinkhorn Optimal Configurations

| Backbone | Step | Losses | Sinkhorn | SMSE | FID |
|---|---|---|---|---|---|
| **Sleep-EDF Dataset** | | | | | |
| Transformer | 25K | $\mathcal{L}_{1.5}, \mathcal{L}_1$ | 3072.97 | 1.58e-3 | 0.8347 |
| Mamba | 100K | $\mathcal{L}_1, \mathcal{L}_1$ | 3440.56 | **1.26e-3** | **0.5541** |
| Mamformer | 75K | $\mathcal{L}_1, \mathcal{L}_{1.5}$ | **2694.93** | 1.96e-3 | 1.0072 |
| **ISRUC Dataset** | | | | | |
| Transformer | 75K | $\mathcal{L}_1, \mathcal{L}_2$ | **2705.60** | 3.49e-3 | 0.1520 |
| Mamba | 25K | $\mathcal{L}_1, \mathcal{L}_1$ | 3227.46 | **2.37e-3** | **0.0928** |
| Mamformer | 75K | $\mathcal{L}_1, \mathcal{L}_{1.5}$ | 2752.93 | 3.28e-3 | 0.1048 |

(matching the metric space of the data) is a distinct optimization phase that does not always align temporally with the distributional convergence measured by FID or the spectral convergence measured by SMSE.

## D.3. Full ECG Results

We now present an analysis of the ablation study for both ECG datasets, first with the optimal loss configuration for each of the backbones and then an analysis of the convergence behavior across the three metrics studied. Overall, we can state that the performance of our framework is highly sensitive to the interaction between the backbone architecture and the choice of losses for the Structure and Detail models.

### D.3.1. PTBXL Optimal Loss Configuration Across Backbones

- For the **Transformer** backbone, we see that the Structure model performed best with the $\mathcal{L}_1$ loss, while the Detail model did so with both the $\mathcal{L}_{1.5}$ and the $\mathcal{L}_2$ losses. This loss preference suggests that the attention mechanism requires a robust anchor (the MAE loss) to prevent global drifts before refining textures with the SMSE loss.

- The **Mamba** backbone excelled in spectral fidelity (SMSE $4.04 \times 10^{-8}$) using the ($\mathcal{L}_2$, $\mathcal{L}_1$) loss combination, for LF and HF models, respectively. This architecture seems particularly adept in modeling the periodicity of ECG signals, leading to the lowest spectral errors across all backbones.

- Finally, the hybrid **Mamformer** architecture achieved the best FID score (0.0207) with an loss combination of ($\mathcal{L}_2$, $\mathcal{L}_1$), just like the with the Mamba backbone. We can conclude that using a mean-seeking objective for the global structure and a median-seeking objective fir the HF details helps capture sharp transitions without sacrificing the stability of the baseline signals.

- **Convergence:** The metrics exhibit distinct speeds of convergence, providing insights into what the framework learns first. The SMSE converges quickest, reaching near-optimal levels by 50K steps. We see the fastest and most stable spectral convergence with the Mamba backbone, whereas the Transformer and Mamformer architectures see a diminishing quality after the 75K step mark. The Sinkhorn distance metric also stabilizes relatively early in training, reaching a plateau once the general shape of the heartbeats is established. Finally, the FID metric consistently improves with longer training. For the Mamformer backbone, we observed steady improvements of around $20\%$ between successive checkpoints, indicating that the specific semantics of the data require more iterations to fully appear.

In summary, for PTB-XL, the Mamformer with the ($\mathcal{L}_2, \mathcal{L}_1$) loss combination is the superior configuration, and while 50K steps are enough to get the rhythm right, the full 100K steps are necessary to achieve the SoA semantic quality (FID) reported in the paper.

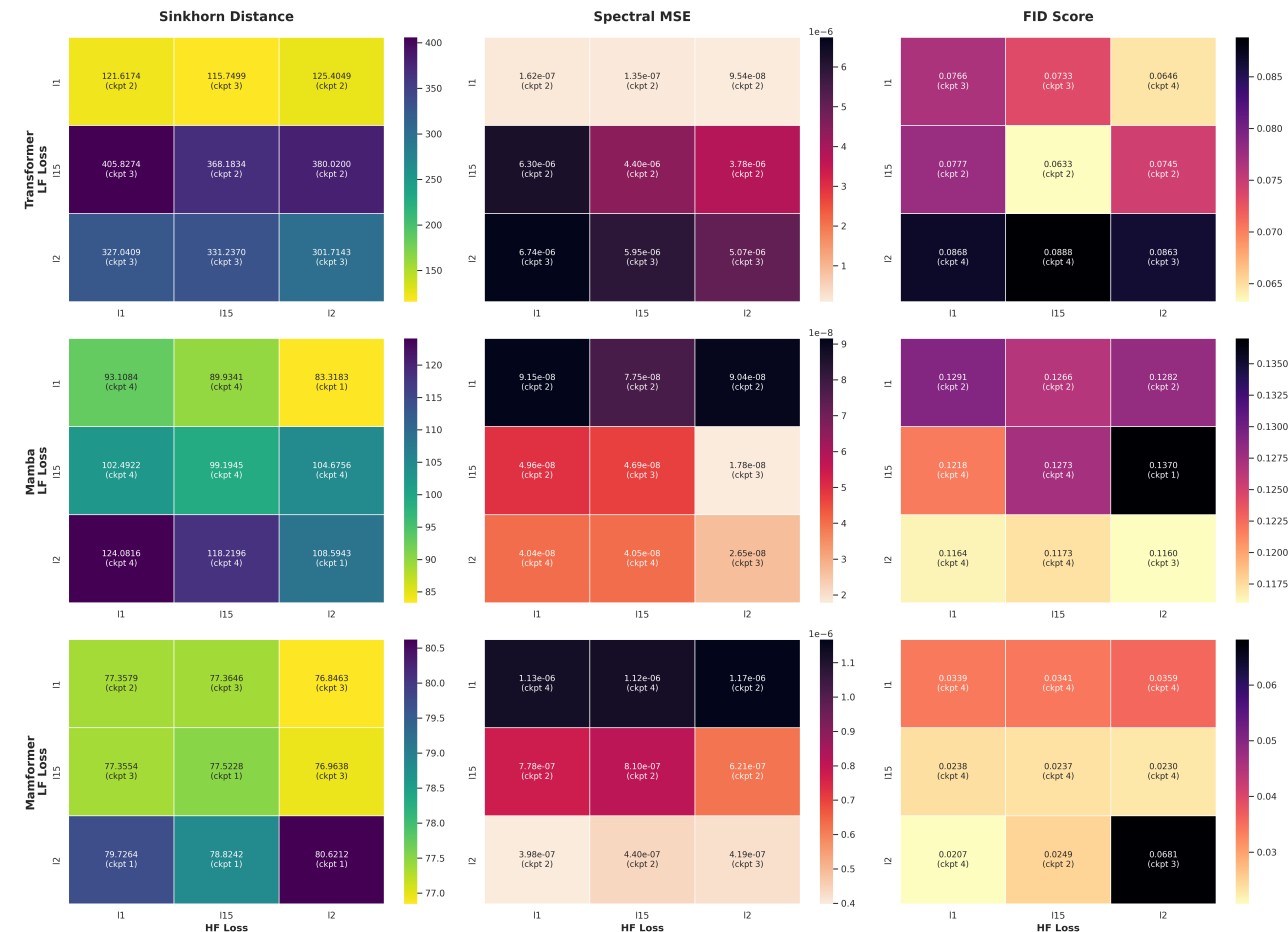

*Figure 9.* PTBXL full ablation results

### D.3.2. Chapman Optimal Loss Configuration Across Backbones

- For the **Transformer** backbone, the best semantic fidelity (FID 0.0574) was achieved with the $\mathcal{L}_{1.5}$ loss for both the Structure and Detail models. This suggests that for the Chapman dataset, which exhibits higher variance in morphology than PTB-XL, the balanced $p$-value of 1.5 provides the necessary flexibility to capture signal trends without the extreme rigidity of the MAE ($\mathcal{L}_1$) or the excessive smoothing of the MSE ($\mathcal{L}_2$).

- The **Mamba** backbone again demonstrated superior spectral modeling, reaching its lowest spectral error (SMSE $6.06 \times 10^{-8}$) very early in training (25K steps) with the ($\mathcal{L}_2$, $\mathcal{L}_1$) loss combination. While it excelled at capturing the frequency distribution, its semantic fidelity (FID 0.1205) lagged behind the other backbones, indicating that while SSMs are excellent in learning oscillations, they may struggle with the specific cross-channel semantic for the FID metric.

- The hybrid **Mamformer** architecture proved to be the top performer for this dataset, reaching the best overall FID (0.0329) using an ($\mathcal{L}_{1.5}$, $\mathcal{L}_1$) configuration. The combination of $\mathcal{L}_{1.5}$ for the global structure and $\mathcal{L}_1$ for the high-frequency details effectively captures the nature of the Chapman signals while maintaining a stable morphological skeleton.

- **Convergence:** Similar to PTB-XL, we observe a hierarchy of convergence speeds. The SMSE reaches optimal or near-optimal levels almost immediately, particularly in the Mamba and Transformer backbones. The Sinkhorn distance (measuring geometric alignment) stabilizes relatively early, with the Mamformer maintaining the most consistent geometric scores (around 200) throughout training. The FID metric, however, benefits significantly from longer training: for the Transformer, we saw a decrease of over $50\%$ between 25K and 100K steps.

In summary, for the Chapman dataset, the Mamformer with $\mathcal{L}_{1.5}/\mathcal{L}_1$ loss is the superior configuration. While the model captures the correct spectral components within the first 25K-50K steps, the continued refinement of the HF details in the latter half of training is essential for minimizing semantic distance (FID).

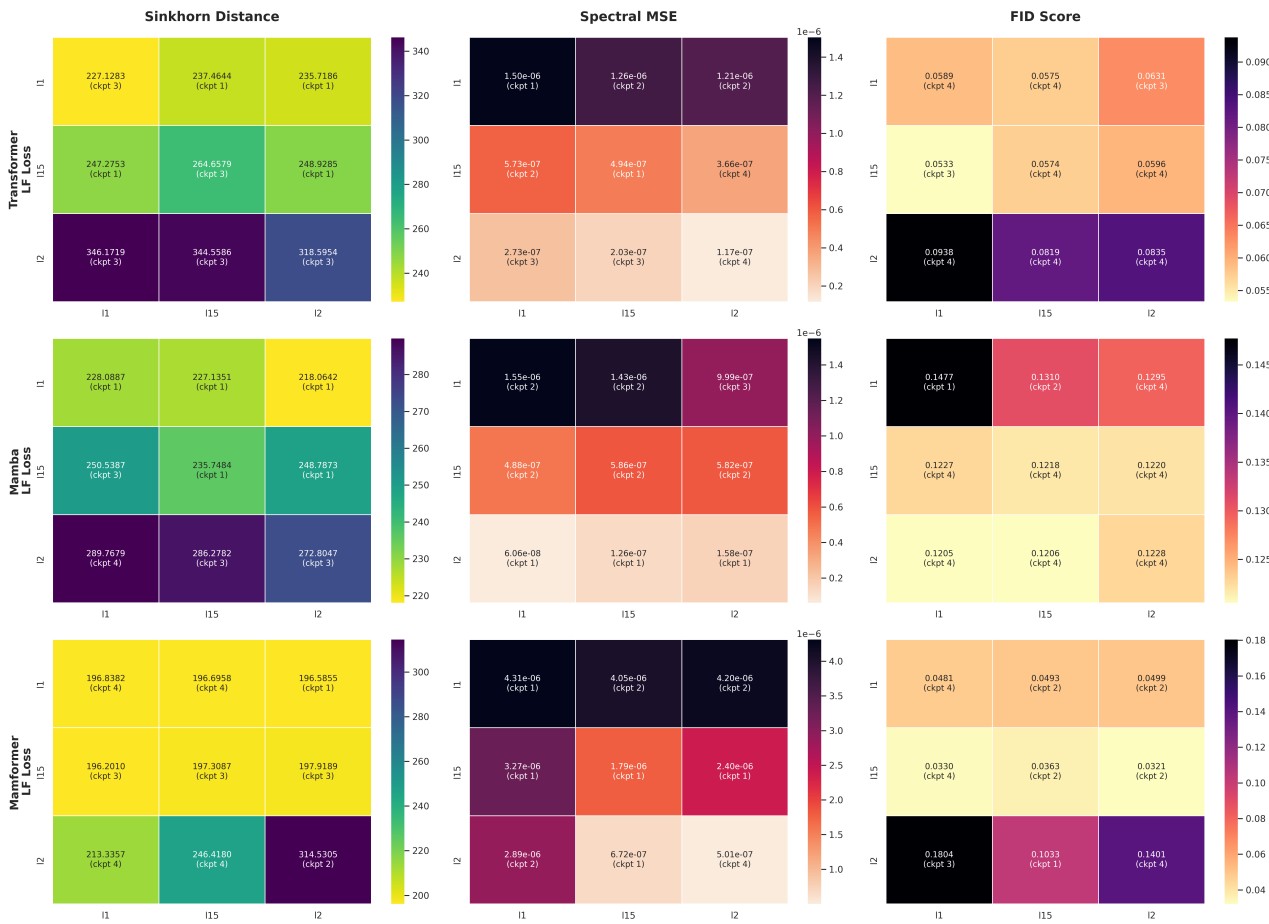

*Figure 10.* Chapman full ablation results

## D.4. Full EEG Results

Here we present an analysis of the ablation study for both EEG datasets, following the same format as before. Again, for the more stochastic EEG signals, our framework proved to be highly sensitive to the choice of backbone architecture and loss combination.

### D.4.1. ISRUC OPTIMAL LOSS CONFIGURATION ACROSS BACKBONES

- The **Transformer** backbone demonstrated its best semantic fidelity with an $(\mathcal{L}_2, \mathcal{L}_1)$ configuration, achieving an FID of 0.0705. While this setup provided solid semantic results, it required longer training to stabilize compared to other architectures.

- The **Mamba** backbone was again the leader in spectral fidelity, achieving a remarkably low SMSE of $3.63 \times 10^{-6}$ with the usual $(\mathcal{L}_2, \mathcal{L}_1)$ loss pairing. Its ability to model the HF components of sleep EEG (such as spindles and beta waves) is evident in its rapid spectral convergence, although its geometric alignment (Sinkhorn) lagged behind the hybrid model.

- The **Mamformer** architecture achieved the best overall performance on this dataset, reaching the best FID of 0.0329 with the loss configuration of $(\mathcal{L}_{1.5}, \mathcal{L}_1)$. This reinforces the trend seen in the Chapman dataset: the $\mathcal{L}_{1.5}$ loss provides

the optimal trade-off between baseline stability and morphological flexibility for the Structure model.

- **Convergence:** The convergence dynamics on ISRUC highlight the challenge of modeling stochastic EEG data. Unlike ECG, where spectral metrics stabilize instantly, the SMSE for ISRUC required more steps to reach their minimum, particularly for the Transformer. The FID metric showed a strong dependency on training duration: for the best-performing Mamformer model, FID improved by approximately $29\%$ between the first and final checkpoints. This suggests that the model first learns the dominant Delta/Theta waves (LF) then gradually the transient events (HF) that define specific sleep stages.

In summary, for the ISRUC dataset, the Mamformer with $(\mathcal{L}_{1.5}, \mathcal{L}_1)$ losses is the definitive choice. The results underscore the necessity of the Detail model's specialized training, as the high-frequency spectral content of sleep EEG data is too complex to be captured by a the traditional loss functions alone.

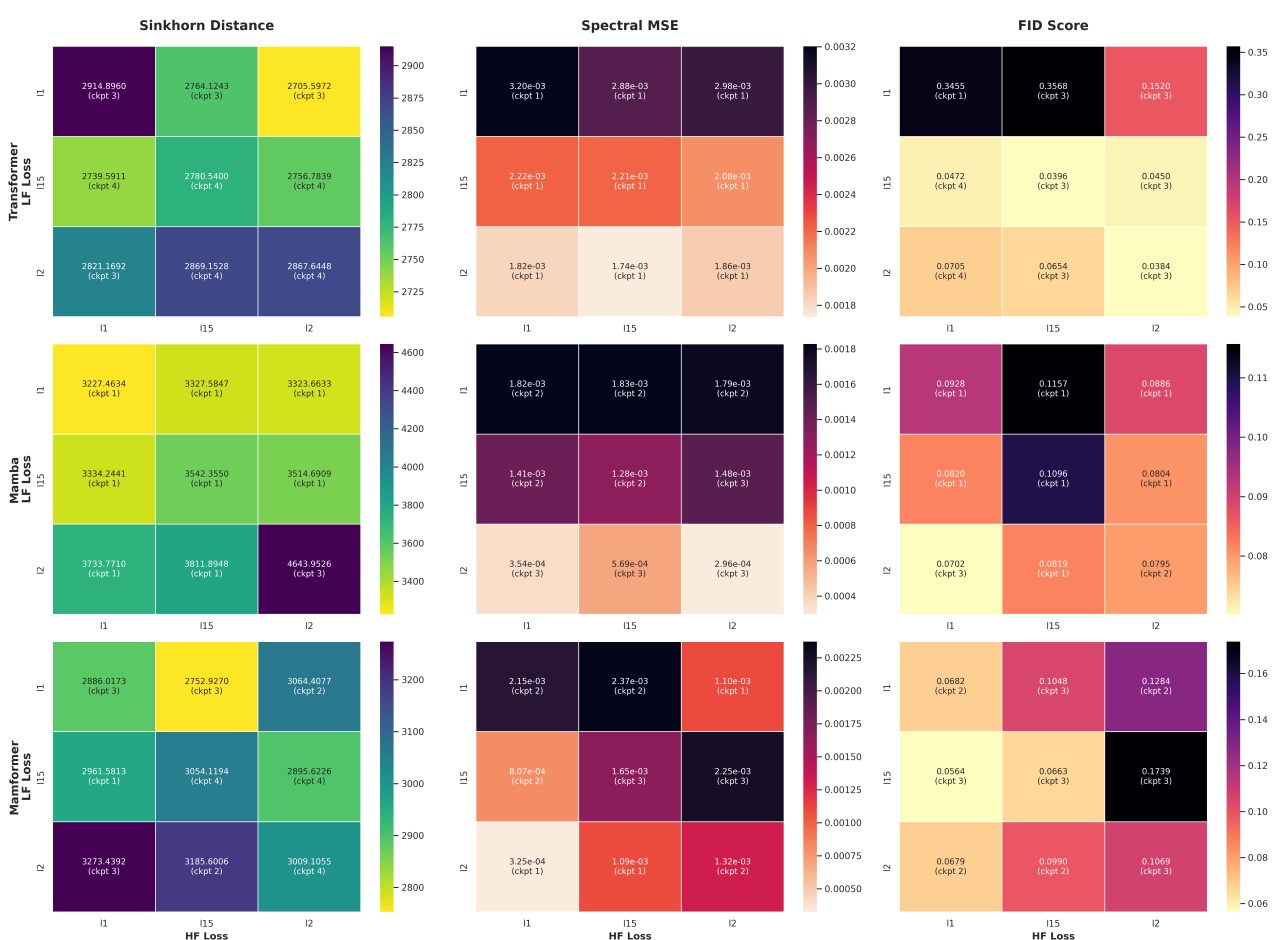

*Figure 11.* ISRUC full ablation results

### D.4.2. SLEEP-EDF OPTIMAL LOSS CONFIGURATION ACROSS BACKBONES

- The **Transformer** backbone struggled the most with this high-entropy dataset. Its best semantic fidelity (FID 1.8255) was achieved with an $(\mathcal{L}_2$ (LF), $\mathcal{L}_1)$ configuration, but it failed to converge to a competitive level compared to the SSM-based architectures. This reinforces the limitation of standard attention mechanisms in modeling long-range, stochastic dependencies without excessive computational overhead.

- The **Mamba** backbone excelled in spectral accuracy, achieving an SMSE of $8.28 \times 10^{-4}$ with the $(\mathcal{L}_2, \mathcal{L}_1)$ loss pair. Its ability to capture the rapid oscillatory transitions of sleep stages (e.g., K-complexes) was superior to the Transformer, though it still exhibited some geometric misalignment (Sinkhorn 3757.1).

- Once again, the **Mamformer** architecture proved to be the optimal choice, achieving a best-in-class FID of 0.3805 with the $(\mathcal{L}_2, \mathcal{L}_1)$ configuration. It successfully balanced the structural consistency required for sleep staging with the high-frequency textural generation needed for realistic EEG synthesis.

- **Convergence:** The convergence profile for Sleep-EDF was distinct. As wit the other EEG dataset, metrics showed continuous improvement throughout training. The FID for the Mamformer model started at 0.3805 (already lower than the Transformer's best) and maintained this lead. The SMSE values were generally higher than for ECG, reflecting the inherent stochasticity and $1/f$ noise characteristics of EEG data that are harder to minimize than quasi-periodic cardiac signals.

In summary, for Sleep-EDF, the Mamformer with $(\mathcal{L}_2, \mathcal{L}_1)$ losses is the superior configuration. The results highlight that while pure SSMs (Mamba) are excellent spectral modelers, the hybrid attention-SSM approach (Mamformer) is necessary to capture the full semantic complexity of sleep EEG patterns.

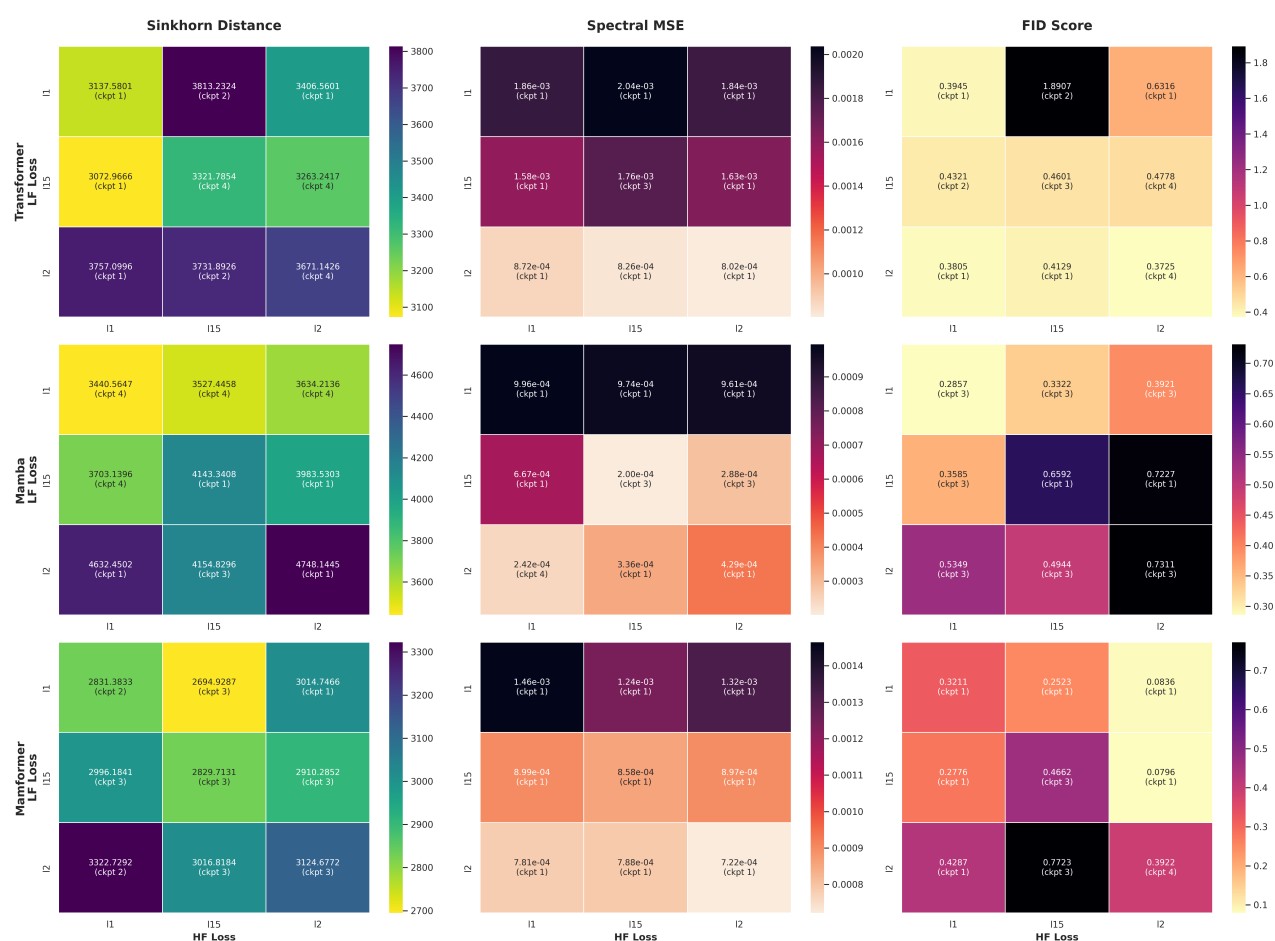

*Figure 12.* Sleep-EDF full ablation results

## D.5. Spectral Analysis of the Best SMSE Models for each Backbone

A key goal in our work is to tackle the spectral shortcomings of Generative Modeling frameworks (see Section 2.2). Here we include the PSD plots of the real data compared to the best-performing model (with respect to loss combination) for each backbone architecture, similar to the plots in Section B. Note that the **ckpt** indicated in each of the backbones refers to the training step at which convergence was reached for the SMSE metric, these being (25K, 50K, 75K, 100K). We can see in Figures 13, 14, and 15 that the generated data follows closely the spectral plots of the real ECG data, while in Figure 16 we see that there is still a spectral gap present in the synthetic data. We attribute this to the stochasticity of the EEG data, and

the inter-subject variability demands a more complex generation scheme, such as incorporating metadata features as an additional conditioning factor.

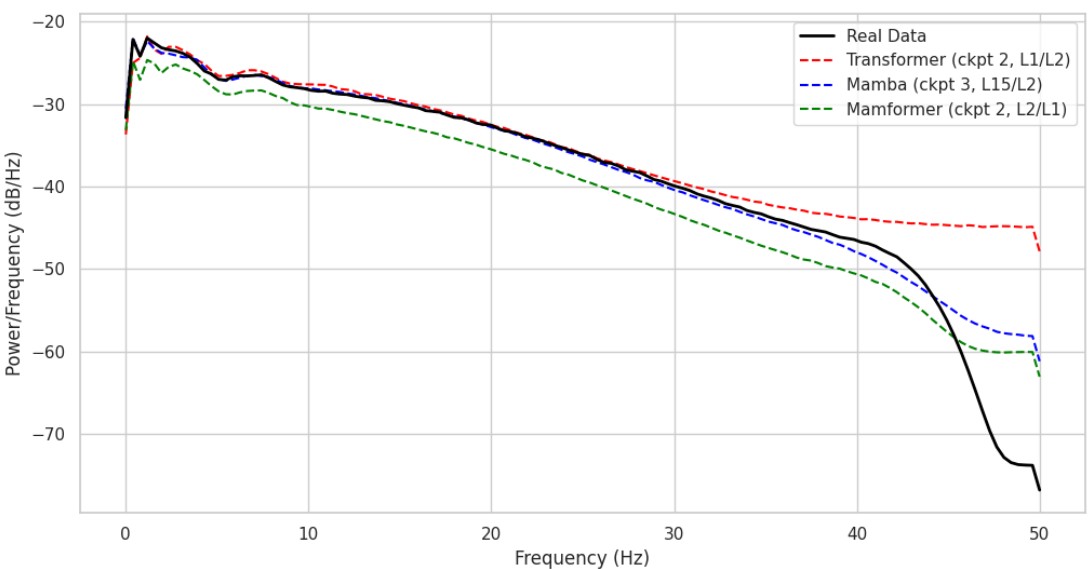

*Figure 13.* PTBXL spectral comparison of best model for each backbone

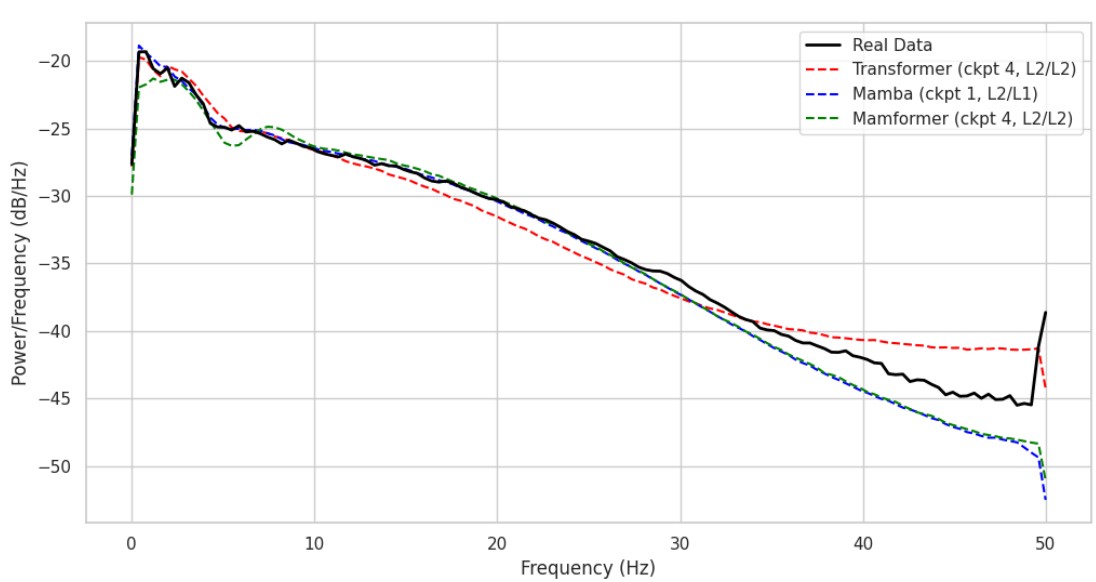

*Figure 14.* Chapman spectral comparison of best model for each backbone

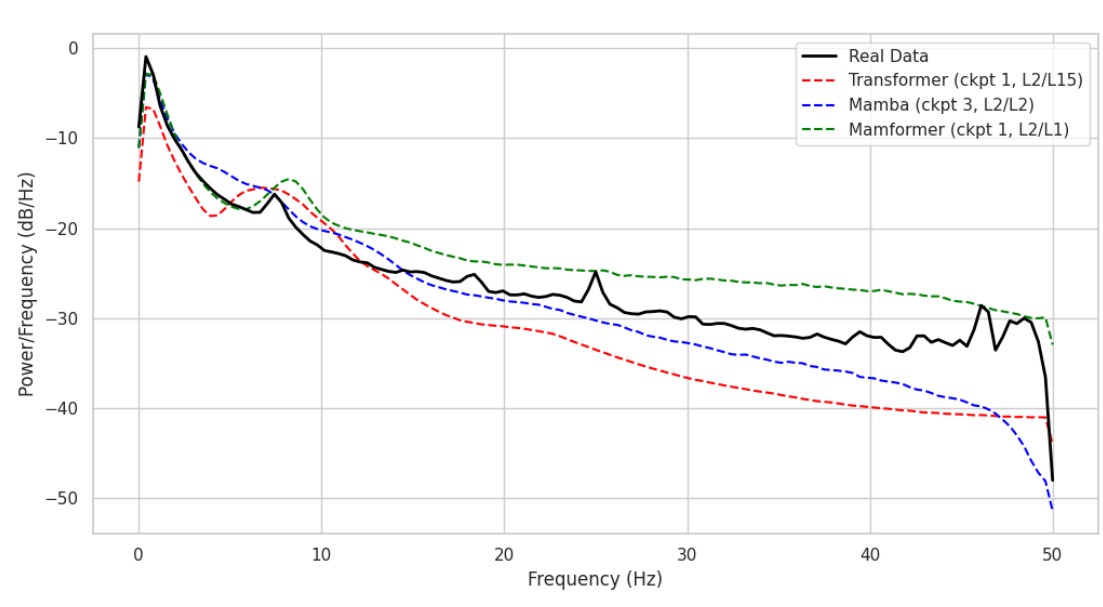

*Figure 15.* ISRUC Spectral comparison of best model for each backbone

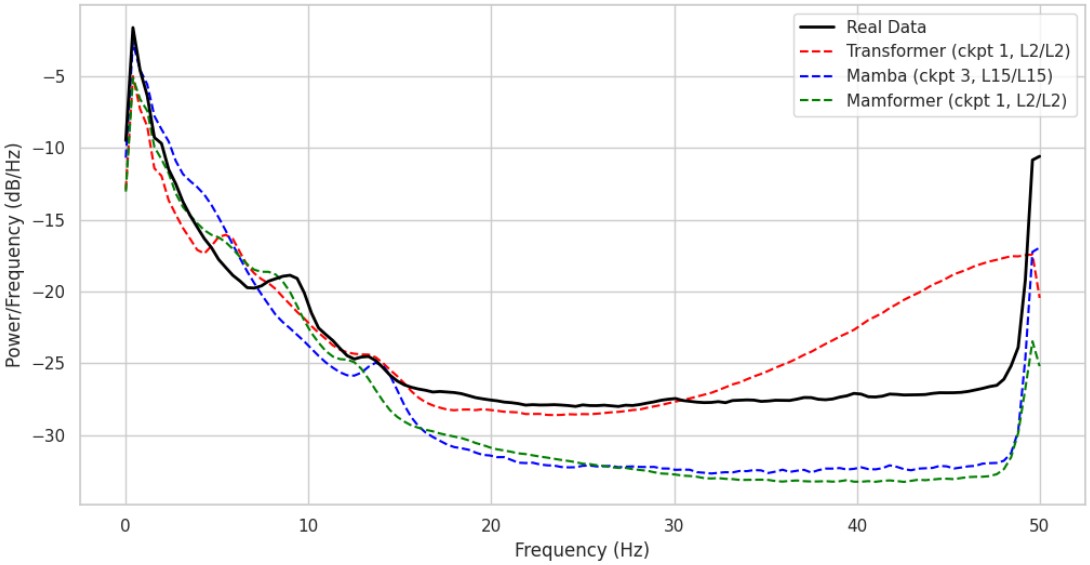

*Figure 16.* Sleep-EDF spectral comparison of best model for each backbone

## Impact Statement

This paper presents a method for high-fidelity biosignal synthesis and super-resolution, aiming to improve the quality of data available for medical machine learning. Our work has the potential to impact societal health outcomes by enabling more efficient data transmission for wearable devices and providing privacy-preserving synthetic datasets for research. While these advancements facilitate better remote diagnostics, we emphasize that generative reconstruction methods should be validated carefully against clinical standards to ensure that synthetic artifacts do not lead to misdiagnosis.

