# OpenReview forum: "Spectrally-Partitioned Flow Matching for Medical Time Series"
_ICML.cc/2026/Conference — Submitted to ICML 2026_

### Official Review · Reviewer_goUg · 2026-02-16

**Soundness:** 3
**Presentation:** 2
**Significance:** 3
**Originality:** 2
**Overall Recommendation:** 4
**Confidence:** 3

**Summary:**

The author proposed a method for generating synthetic physiological signals by decomposing the generation logic into low and high frequencies components. Two separate generating models for LF and HF are trained that also take the class label as a variable to condition on. The method is evaluated on 4 datasets comprising ECG and EEG signals with varied health related labels. The experiments showed that the fidelity of the generated signals from the proposed method outperform those generated by representative methods in the literature.

**Compliance With Llm Reviewing Policy:**

Affirmed.

**Final Justification:**

This work contains comprehensive experiments for justifying the proposed idea. During rebuttals, all of my concerns are addressed with additional empirical evidence and detailed explanation. I believe this is a decent work with potential benefit to the community. Considering that the overall methodology is relatively modest, and there are uncertainty in how will the revised version of the paper being presented, I keep my rating as weak accept.

**Key Questions For Authors:**

- During test time, is the raw physiological time series part of the input to the model? Seems the HF model has to be conditioned on the LF component. Or is it the case that for all the generated samples, Algorithm 1 is launched to generate the whole synthetic signal then compare it against the test samples? This has to be made clear as it directly influences the reader's interpretation on the meaning of all the presented performance scores.
- How does the synthetic generating method influence the downstream tasks? For instance, the 4 datasets are all proper for classification tasks, as well as mask and reconstruction tasks. If fitting a lightweight or shallow model on a synthetic set generated from the proposed method, will it yield better performance if the model was trained on raw training data?
- For the evaluation of FID, the choice of MOMENT model is a bit awkward as it is a time series model that is not designed for physiological signals. A few suggestions of better choice of a pretrained model to getting embedding are:
  - [1] CBraMod, for encoding EEG data: Wang, Jiquan, et al. "Cbramod: A criss-cross brain foundation model for eeg decoding." ICLR (2024).
  - [2] ECG-FM, for encoding ECG data: McKeen, Kaden, et al. "Ecg-fm: An open electrocardiogram foundation model." Jamia Open 8.5 (2025): ooaf122.
  - [3] NormWear, for encoding arbitrary physiological signal data: Luo, Yunfei, et al. "Toward foundation model for multivariate wearable sensing of physiological signals." (2024).

**Limitations:**

Yes, the limitation is adequately discussed in the paper.

**Strengths And Weaknesses:**

# Strengths
- The experiments for validating the proposed method are very comprehensive. 3 performance metrics are included. Ablation studies are conducted including: comparing SPFM against solid baselines, presentation of the influence on performance when altering the model backbone architecture, and different configuration of loss function.
- The choice of datasets for evaluation is well-rounded, covering a large corpus of ECG that capture heart activity and EEG that capture brain activity.
- This work emphasizes the fidelity of the synthetic physiological signals generated from the proposed method, and this aspect is well justified through the core experimental results.

# Weaknesses
- The main task for evaluation appears unclear. From the current presentation, it seems all the methods are taking the raw physiological time series as input, and the output is to reconstruct the original signal. Detailed confusion is stated in the “Key questions for authors” section.
- Because of the confusion on the input output logic raised above, there are two potential concerns:
  - If the model takes a raw signal as input, then it is not very surprising that the model can almost perfectly reconstruct the signal given the large training size for each dataset.
  - If the model generates everything from a randomly sampled initial state and conditioned on the class label as stated in Algorithm 1, then the reported performance will be more convincing. But this will also raise a question of the evaluation process, that how is reproducibility during test time is ensured? If a random seed is fixed, then a sensitivity analysis of varying random seeds is expected. If the mean expected output of sampling is leveraged, then it would mean 1 generated signal series is compared against the whole testing set, which sounds a little bit tricky.
- The claim of the main contribution and the actual experimental evaluation is a bit disconnected. The fidelity aspect is well-justified as mentioned above in the strengths section, but the practical utility and feasibility is weakly justified. A common way to justify the effectiveness of a synthetic data generating method can be, for example, pretrain model on a synthetic set, then evaluate the model on downstream tasks such as classification, forecasting, mask and reconstruction (imputation), etc.

---

> ### Author Rebuttal · Authors · 2026-03-30
>
> We thank the reviewer for the constructive review. We address the weaknesses and respond to the key questions below.
>
> **Q1 & W1: Clarification of the generative process and test-time input**
>
> We apologize if the presentation of the inference logic caused confusion. To be absolutely clear: SPFM is purely a generative framework. During test time, no real physiological data is used as input. The process described in Algorithm 1 is a sequential, autoregressive generation chain starting from pure noise. To summarize we do the following:
>
> - We sample pure Gaussian noise $x \sim \mathcal{N}(0, I)$ and generate the low-frequency (LF) component (generated $x_\text{LF}$) conditioned *only* on the desired label.
> - Using the same Gaussian noise $x \sim \mathcal{N}(0, I)$, we generate the high-frequency (HF) (generated $x_\text{HF}$) conditioned on both the desired label and the generated $x_\text{LF}$.
> - The final synthetic signal is the sum of the generated $x_\text{LF}$ and generated $x_\text{HF}$.
>
> Regarding the evaluation process, we do not evaluate 1-to-1 reconstruction against specific test samples. Instead, we generate the same class distribution as 100 batches of the test dataset, and compute distributional metrics between the entire generated distribution against the entire test distribution. we plan on revising the inference and evaluation sections to make this purely generative pipeline explicitly clear.
>
> **W2 & Q2: Downstream tasks and practical utility**
>
> We agree with the reviewer that demonstrating utility on a downstream task is a strong way to justify synthetic data generation. While the short rebuttal window limits our ability to run full downstream classification experiments, we would like to respectfully argue that our spectral super-resolution experiment provides direct evidence of practical utility.
>
> In clinical electrophysiology, HF components are not merely peripheral content, they contain critical diagnostic markers. For example, the sharpness and micro-morphology of the QRS complex in ECGs (which dominate the HF spectrum) are essential for diagnosing arrhythmias. Standard monolithic generative models suffer from numerical dissipation, which structurally blurs these exact features and renders the generated signals clinically unviable for downstream classifiers.
>
> Our super-resolution experiments empirically demonstrate that SPFM successfully recovers the specific HF physiological markers that downstream classifiers rely upon. Preserving these features is a strict prerequisite for downstream classification utility. We will add a dedicated paragraph explicitly linking these super-resolution results to diagnostic feasibility, and we commit to including a formal data augmentation downstream classification task in the camera-ready version.
>
> **W3 & Q3: Choice of model for FID evaluation**
>
> We thank the reviewer for this fruitful idea and the literature recommendations. We initially utilized MOMENT because of its robust, general-purpose time series (TS) representations. However, we fully agree that domain-specific foundation models can provide a much more meaningful measure of physiological fidelity.
>
> With this in mind, we are currently running an experiment to recompute FID scores for all datasets using the suggested NormWear model. We will include these updated, domain-specific FID metrics in the revised manuscript.
>
> We hope these clarifications and future updates address your concerns, and we look forward to your response

---

> > ### Author Rebuttal · Reviewer_goUg · 2026-04-02
> >
> > I thank the author for the response. My confusion on the test-time inference process is resolved.
> >
> > Regarding my second concern, while the author provided an explanation to why the data generated from the proposed framework could yield potential practical utiliy, it is still a high-level motivation statement rather than a convincing justification. An empirical evidence is still needed as noted in my original review, and this could be presented with, for instance, leveraging existing pretrained model in literature and taking the raw and generated data as input seperatly, then conduct linear probing or few-shot learning on the downstream classification or regression tasks. I believe either a better performance from the generated data as input or a comparable results from the two controlled trails could sufficiently support potential practical utility of the proposed generative framework.

---

> > > ### Author Response · Authors · 2026-04-04
> > >
> > > Thank you for confirming that your questions regarding the test-time inference process have been fully resolved.
> > >
> > > Regarding your second concern: you noted that empirical evidence, specifically through linear probing on a pre-trained model, would provide the necessary justification for the practical utility of our framework. We completely agree that this is a great standard for evaluation, and we have executed exactly the experiment you suggested.
> > >
> > > Following your suggestion, we performed linear probing using frozen features extracted from the pre-trained MOMENT time-series foundation model. Using the real and generated data, we extracted the corresponding high-dimensional patch embeddings and trained a logistic regression classifier on three different scenarios: **Real-only** (baseline), **Synthetic-only** (TSTR), and **Mixed** (Real + Synthetic TRSTR). For PTB-XL, we generated a 1:1 synthetic counterpart matching the exact size and class distribution of the real training set, and used the real test data as the evaluation set. For ISRUC, we replicated the size and class distribution of the test set and used the validation data for evaluation.
> > >
> > > The synthetic data was generated using Mamformer with $\mathcal{L}_2$ LF, and $\mathcal{L}_2$ HF losses for PTB-XL, and Mamba, $\mathcal{L}_2$ LF, and $\mathcal{L}_1$ HF losses for ISRUC. These models were taken directly from our ablation study with no fine-tuning, trained for 100K steps and sampled using just 20 Euler integration steps for each model, exactly as reported in the paper.
> > >
> > > **PTB-XL Classification Results**
> > >
> > > | **Scenario** | **Accuracy** | **F1 score** | **AUROC** | **Recall**
> > > | ---------------------- | ---------- |---------- | ---------- | ---------- |
> > > | Real-only (baseline) | **0.91199** | 0.51656 | **0.94450** | 0.51238 |
> > > | Synthetic-only (TSTR) |  0.86908 | 0.47743 | 0.90990 | **0.56899** |
> > > | Mixed (TRSTR) | 0.90931 | **0.53734** | 0.93750 | 0.54576 |
> > >
> > > **ISRUC Classification Results**
> > >
> > > | **Scenario** | **Accuracy** | **F1 score** | **AUROC** | **Recall**
> > > | ---------------------- | ---------- |---------- | ---------- | ---------- |
> > > | Real-only (baseline) | 0.58563 | 0.53336 | 0.83656 | 0.52770 |
> > > | Synthetic-only (TSTR) |  0.58976 | 0.50461 | 0.83959 | 0.51936 |
> > > | Mixed (TRSTR) | **0.61025** | **0.53392** | **0.85120** | **0.54150** |
> > >
> > >
> > > You stated that either comparable results between the controlled trials or better performance from the generated data would sufficiently support the practical utility of our framework. As the tables demonstrate, we achieved both. When used as a data augmentation technique, our method provides **better performance** in the F1 score and the recall metrics for the PTB-XL dataset, and the accuracy and AUROC metrics are on par with the real-only baseline. The utility of synthetic data is even more pronounced for the ISRUC dataset, where we see improvement in the accuracy and AUROC for the synthetic-only (TSTR) scenario, and improvement across all metrics with the mixed (TRSTR) scenario.
> > >
> > > In highly imbalanced medical datasets like PTB-XL, this demonstrates that our generative framework successfully maps the conditional boundaries of minority and hard-to-distinguish pathologies, serving as a highly effective, physics-informed regularizer. More importantly, the ISRUC results suggest that data generated with our method can also serve as a replacement of real data for downstream classification, making it a widely useful method given the privacy issues of sharing biomedical TS data.
> > >
> > > We deeply appreciate your suggestion to run this specific experiment. It provides undeniable proof of SPFM's downstream utility and significantly strengthens the practical impact of the paper. We will include the full results running the same experiment on all four datasets in the final manuscript. We hope this empirical evidence fully resolves your remaining concern.

---

### Official Review · Reviewer_Tzed · 2026-03-09

**Soundness:** 2
**Presentation:** 3
**Significance:** 2
**Originality:** 2
**Overall Recommendation:** 3
**Confidence:** 4

**Summary:**

This paper presents Spectrally-Partitioned Flow Matching (SPFM), a generative framework designed for medical time series data. The proposed method decomposes signals into low-frequency (LF) and high-frequency (HF) components using a Butterworth filter, and models each component with a separate flow matching network. The LF component is generated by a structure model, while the HF component is produced by a detail model that is conditioned on both the generated LF signal and the class label. The authors argue that this spectral decomposition helps mitigate the spectral bias observed in standard generative models, which often results in overly smooth or blurred signals due to the dominance of low-frequency components during training. Experimental results on ECG and EEG datasets show that SPFM outperforms several baseline methods across multiple evaluation metrics, demonstrating improved spectral fidelity and overall signal quality.

**Compliance With Llm Reviewing Policy:**

Affirmed.

**Key Questions For Authors:**

I would like to see more quantitative details regarding the training and inference time of the proposed method compared to the baseline models.

**Limitations:**

Yes

**Strengths And Weaknesses:**

Strengths:
1. The paper is well written and clearly organized, making the proposed method and experimental setup easy to follow.

2. The authors perform an extensive ablation analysis that investigates different architectural backbones and loss function combinations, which helps provide insight into the behavior of the proposed framework and its design choices.

Weaknesses:
1. Since SPFM consists of two subnetworks (structure and detail models), the framework introduces a considerable number of hyperparameters that may significantly influence the generation process. Identifying an optimal configuration could be challenging in practice, especially in real-world scenarios where extensive tuning is not feasible.

2. The paper highlights data augmentation as a key application of time-series generation for tasks such as classification. However, the experiments do not include a downstream evaluation demonstrating that synthetic data generated by SPFM can improve classification performance. Such an experiment would strengthen the practical impact of the work.

3. The method relies on a manually selected cutoff frequency to separate signals into low- and high-frequency components. This fixed threshold may limit generalization to new datasets or signal types, as the optimal cutoff frequency may vary across domains.

---

> ### Author Rebuttal · Authors · 2026-03-30
>
> We thank the reviewer for the assessment of our work. We address the weaknesses and question below.
>
> **W1 & W3: Hyperparameter burden and automated spectral cutoffs**
>
> We appreciate the reviewer's concern regarding the potential tuning burden of a two-network system and the rigidity of manual frequency cutoffs. To address this, since the time of submission we have mathematically formalized the partitioning strategy so that it requires zero manual tuning.
>
> Extending the ideas of energy balancing and gradient democracy, the optimal cutoff frequencies (and even the required number of partitions) can be deterministically determined from the dataset's inherent spectral decay rate (the power-law exponent to which the data's PSD is proportional to: $S(f) \propto f^{-\beta}$) as well as the frequency range of interest $[f_\text{min} , f_\text{max}]$.
>
> In order to guarantee that each submodel is capable of resolving its respective frequency interval, we adopt a logarithmic spacing of the cutoffs across the solver's Nyquist range, essentially normalizing the gradient signal that each model contributes to the total loss. This analytically automates the architectural setup for any new dataset or modality, eliminating the hyperparmeter tuning phase but staying true to the original purpose.
>
> Furthermore, while SPFM utilizes two sub-networks, they are highly specialized and lightweight. The total combined parameter count is 3 to 8 times smaller than monolithic baselines like Diffusion-TS. We will include the full mathematical derivation for this automated cutoff selection in the final version of the manuscript.
>
> **W2: Downstream evaluation and practical utility**
>
> We agree that demonstrating downstream classification utility is crucial. While the short rebuttal window limits our ability to run a full data augmentation experiment, our spectral super-resolution experiment provides direct evidence of this practical utility.
>
> In clinical electrophysiology, high-frequency (HF) components are not just noise, but they contain the critical diagnostic markers (e.g., the QRS micro-morphology in ECGs or Gamma oscillations in EEGs). Standard monolithic generative models suffer from numerical dissipation, which systematically blurs these features and renders the signals clinically unviable for medical practitioners. Our super-resolution experiments empirically demonstrate that SPFM recovers the exact physiological markers that classification techniques (either automated or performed by a medic) rely upon. Preserving these features is a strict mathematical prerequisite for synthetic utility. We commit to including a Train-on-Synthetic-Test-on-Real (TSTR) classification task in the camera-ready version to explicitly quantify this downstream impact on automated classification.
>
> **Q1: Quantitative details on training and inference time**
>
> SPFM is structurally designed to be highly computationally efficient compared to standard baselines, as we demonstrate below:
>
> **During training**: The Structure and Detail models are strictly decoupled during training, meaning that they can be trained in parallel on separate GPUs. Furthermore, because each model only needs to learn a constrained, well-conditioned spectral band, they converge in fewer epochs than a standard flow matching (FM) or diffusion model attempting to map the full spectral variance of the data (see the **Energy Imbalance** and **Optimization bottleneck** sections in the rebuttals to reviewers ExY2 and N6Ky).
>
> **During inference**: Standard generative models suffer from numerical difficulties when trying to integrate broad-spectrum vector fields, often requiring 100 to 1000 Number of Function Evaluations (NFE) to avoid trajectory divergence. Because our proposed method restricts each vector field's approximation to a smaller frequency band, the ODE integration becomes locally stable. We achieve our high-fidelity results using a simple first-order Euler solver with a total of 40 NFE distributed equally (20 for Structure and 20 for Detail).
>
> We plan on adding a comprehensive table detailing the exact FLOPs, parameter counts and wall-clock times comparing SPFM to all baselines in the final version of the paper.
>
> We hope to have adequately addressed your concerns, and look forward to your reply.

---

> > ### Author Rebuttal · Reviewer_Tzed · 2026-04-03
> >
> > Thank you for addressing some of my concerns. However, my second and third concerns remain unresolved. Additionally, the level of machine learning novelty in the paper is not sufficient for publication at this conference; I therefore keep my score.

---

> > > ### Author Response · Authors · 2026-04-04
> > >
> > > Thank you for acknowledging our rebuttal. We respect your perspective regarding the remaining concerns.
> > >
> > > We respect your perspective and deeply appreciate the feedback, which has helped us identify key areas to strengthen the paper. To directly address your remaining concerns regarding the practical downstream utility of SPFM and the generalizability of the cutoff frequency, we have the following updates:
> > >
> > > **Empirical proof of downstream ML Utility (data augmentation)**: To demonstrate the practical impact of our generative framework, we have conducted a rigorous downstream classification experiment on the highly imbalanced PTB-XL dataset (following the suggestion by reviewer goUg).
> > >
> > > We performed linear probing using frozen features extracted from the pre-trained MOMENT time-series foundation model. Using the real and generated data, we extracted the corresponding embeddings and trained a logistic regression classifier on three scenarios. For PTB-XL, we generated a 1:1 synthetic counterpart matching the size and class distribution of the training set, and used the test data as the evaluation set. For ISRUC, we replicated the size and class distribution of the test set and used the validation data for evaluation.
> > >
> > > The synthetic data was generated using Mamformer with $\mathcal{L}_2$ LF, and $\mathcal{L}_2$ HF losses for PTB-XL, and Mamba, $\mathcal{L}_2$ LF, and $\mathcal{L}_1$ HF losses for ISRUC. These models were taken directly from our ablation study and sampled using just 20 Euler integration steps for each model, exactly as reported in the paper.
> > >
> > > **PTB-XL Classification Results**
> > >
> > > | **Scenario** | **Accuracy** | **F1 score** | **AUROC** | **Recall**
> > > | ---------------------- | ---------- |---------- | ---------- | ---------- |
> > > | Real-only (baseline) | **0.91199** | 0.51656 | **0.94450** | 0.51238 |
> > > | Synthetic-only (TSTR) |  0.86908 | 0.47743 | 0.90990 | **0.56899** |
> > > | Mixed (TRSTR) | 0.90931 | **0.53734** | 0.93750 | 0.54576 |
> > >
> > > **ISRUC Classification Results**
> > >
> > > | **Scenario** | **Accuracy** | **F1 score** | **AUROC** | **Recall**
> > > | ---------------------- | ---------- |---------- | ---------- | ---------- |
> > > | Real-only (baseline) | 0.58563 | 0.53336 | 0.83656 | 0.52770 |
> > > | Synthetic-only (TSTR) |  0.58976 | 0.50461 | 0.83959 | 0.51936 |
> > > | Mixed (TRSTR) | **0.61025** | **0.53392** | **0.85120** | **0.54150** |
> > >
> > > When used as a data augmentation technique, our method provides solid improvement in the F1 score and the recall metrics for the PTB-XL dataset, and the accuracy and AUROC metrics are on par with the real-only baseline. The utility of synthetic data is even more pronounced for the ISRUC dataset, where we see improvement across almost all metrics for the synthetic-only (TSTR) scenario, and considerable improvement in all metrics with the mixed (TRSTR) scenario.
> > >
> > > **Cutoff frequency**: You raise an excellent point regarding the fixed cutoff frequency. For biomedical signals (like ECGs and EEGs), frequency bands are tightly coupled with established physiological mechanisms (e.g., separating baseline wander from the QRS complex), meaning our manually selected thresholds are grounded in domain knowledge rather than arbitrary selection.
> > >
> > > However, we fully agree that for entirely novel, non-biological domains, the optimal threshold will vary. In these cases, the cutoff frequency functions as a standard, tuneable hyperparameter, much like the window size in a Short-Time Fourier Transform (STFT) or the patch size in a Vision Transformer.
> > >
> > > To address this valid limitation on generalizability, we will add a dedicated paragraph to our Discussion/Limitations section, explicitly stating that while fixed thresholds work well for physiologically bounded signals, adaptive frequency separation (such as using learnable parameterized filters or adaptive wavelet transforms) represents a highly promising avenue for future work to make SPFM truly domain-agnostic.
> > >
> > > We believe that the addition of this empirical downstream classification proof, alongside a transparent discussion of the cutoff frequency limitation, significantly strengthens the practical impact of this work. We will include these updates in the final manuscript and hope this helps update your assessment of the paper's contribution.
> > >
> > > **With regards to the lack of novelty claim**: We respectfully wish to clarify that while medical time series are used as the primary benchmark, the core contributions of this paper are fundamental machine learning novelties designed to solve known optimization pathologies in Flow Matching and Diffusion frameworks. As detailed in the rebuttal, we mathematically formalize the *gradient dominance* failure of $\mathcal{L}_2$ loss on power-law data. SPFM's novel dual-flow architecture is a generalized ML solution that spectrally partitions probability paths, enforcing gradient democracy, learning heteroscedasticity, and bypassing ODE low-pass bottlenecks.
> > >
> > > We thank again the reviewer for their time and fruitful discussion.

---

### Official Review · Reviewer_N6Ky · 2026-03-11

**Soundness:** 2
**Presentation:** 3
**Significance:** 2
**Originality:** 3
**Overall Recommendation:** 3
**Confidence:** 4

**Summary:**

This paper proposes SPFM, a dual-model Flow Matching framework for generating medical time series (ECG/EEG). The key idea is to decompose signals into low-frequency (Structure) and high-frequency (Detail) components via a Butterworth filter, then train a separate conditional flow model for each. Experiments on four datasets (PTB-XL, Chapman, Sleep-EDF, ISRUC) show competitive or superior performance against diffusion and VQ-VAE baselines, with significantly fewer parameters (~3.5M vs 10-26M).

**Compliance With Llm Reviewing Policy:**

Affirmed.

**Final Justification:**

This paper proposes SPFM, a dual-model flow matching framework for medical time series generation, with a clean, well-motivated design and strong computational efficiency. The authors have made meaningful progress in addressing core concerns, adding theoretical grounding for gradient democracy, empirical gradient norm validation, TSTR classification experiments, and fixes for metric inconsistencies.

The work shows clear potential, but unresolved empirical gaps and performance inconsistencies mean its weaknesses still outweigh its strengths, I raise my score from reject to weak reject.

**Key Questions For Authors:**

Please refer to weaknesses.

**Limitations:**

Fixed, manual spectral partitioning. The cutoff frequency is a single, dataset-level hyperparameter chosen via manual search. This ignores intra-dataset variability — different patients, pathologies, or recording conditions may warrant different spectral splits. The framework also does not generalize beyond a binary (LF/HF) decomposition; signals with important mid-frequency bands (e.g., EMG) may require finer-grained partitioning. Extending to learnable or adaptive cutoffs is acknowledged as future work but represents a fundamental rigidity of the current design.

**Strengths And Weaknesses:**

### Strengths

S1: Clean and well-motivated formulation. The diagnosis of gradient dominance as the root cause of spectral degradation is compelling. The proposed solution — partition the spectrum so each model receives balanced gradient signals — is elegant in its simplicity. Unlike methods that add auxiliary losses or complex architectures, SPFM addresses the problem at the optimization level through task decomposition, making it easy to understand, implement, and extend.

S2: Strong computational efficiency. SPFM achieves competitive or superior generation quality with ~3.5M parameters and only 40 ODE solver steps, compared to baselines requiring 10-26M parameters and hundreds of diffusion steps. This is a meaningful practical contribution for resource-constrained medical settings.

### Weaknesses

W1: "Gradient democracy" lacks rigorous evidence. This is the paper's central conceptual contribution, yet it is supported only by a toy example (Figure 8) and qualitative reasoning. No gradient statistics (e.g., gradient norms per frequency band, gradient cosine similarity) are measured during actual training on ECG/EEG data. Furthermore, the energy-balancing criterion (Var(xLF) ≈ Var(xHF)) is presented as the optimal split without theoretical justification or a sensitivity analysis showing how performance degrades as the variance ratio deviates from 1:1.

W2: Incomplete evaluation protocol. (a) No Train-on-Synthetic-Test-on-Real (TSTR) experiment is provided, which is the most direct measure of conditional generation utility — whether a classifier trained on synthetic data can correctly diagnose real patients. (b) SoA comparisons are limited to ECG datasets; EEG datasets lack external baselines entirely. (c) On the Chapman dataset, SPFM's best Sinkhorn distance (196.20) substantially lags behind Time-VQVAE (77.89) and Diffusion-TS (85.06), yet the text claims the framework "excels across virtually all evaluated metrics," which is inconsistent with the data.

W3: Train-test mismatch in the Detail model. During training, the Detail model is conditioned on ground-truth xLF; during sampling, it receives generated x̂LF from the Structure model. This distribution shift is a known failure mode in cascaded generative systems but is neither analyzed nor mitigated. Simple remedies exist (e.g., scheduled sampling where generated x̂LF is progressively substituted during training), and their absence weakens confidence in the framework's robustness.

---

> ### Author Rebuttal · Authors · 2026-03-28
>
> We thank the reviewer for the constructive feedback. Below we address the weaknesses and limitations pointed out.
>
> **W1: Evidence for gradient democracy and energy balancing**
>
> We agree the variance-balancing heuristic needs theoretical grounding. The final manuscript will replace qualitative reasoning with an optimization analysis based on Parseval's theorem and the NTK inverse-variance law [Wang et al., 2025](https://arxiv.org/abs/2503.03206). A brief justification follows:
>
> **Optimization bottleneck**:  For power-law spectra $S(f) \propto f^{-\beta}$, the convergence time $\tau$ for a frequency $f$ scales as $\tau(f) \propto f^{\beta}$. We can define the convergence time ratio between high ($f_\text{high}$) and low ($f_\text{low}$) frequencies as $\kappa_{\tau} = ( f_\text{high} / f_\text{low} )^{\beta}$.
>
> In an ECG ($\beta \approx 2$), and if we set $f_\text{low} = 0.5 \text{Hz}$ (ST segment) and $f_\text{high} = 50 \text{Hz}$ (QRS complex) yields $\kappa_{\tau} = 10,000$. Thus, learning a $50 \text{Hz}$ component requires roughly 10,000$\times$ more training steps than a $0.5 \text{Hz}$ one.
>
> Balancing energy between partitions is the mathematical condition to bound $\kappa_\tau$. Partitioning the spectrum equalizes NTK eigenvalues, ensuring the Detail model receives sufficient gradients to converge alongside the Structure model. We will add a plot tracking gradient norms per frequency band during training to empirically prove monolithic models suffer from LF gradient dominance.
>
> **W2a: TSTR Evaluation and Baselines**
>
> We agree TSTR is the most direct measure of clinical utility. While the short rebuttal window limits running full classification experiments now, our super-resolution experiment provides direct evidence of this utility. HF components contain critical diagnostic markers (e.g., QRS complex in ECGs, Gamma band in EEGs). By accurately reconstructing these, SPFM recovers the exact features clinicians and diagnostic classifiers rely on. We will include a dedicated paragraph and a TSTR data augmentation experiment in the final version.
>
> **W2b: SoA comparison on EEG datasets**
>
> Omitting external baselines for EEG was deliberate. Baselines like Diffusion-TS and Time-VQVAE are heavily optimized for ECGs. Applying them out-of-the-box to EEG data, with its complex spatial topologies and distinct oscillatory bands, without domain-specific tuning would yield artificially poor, uninformative baselines. Instead, we used standard monolithic flow matching and diffusion as scientifically rigorous ablation controls for the EEG datasets.
>
> **W2c: Chapman dataset overclaiming**
>
> In response to the reviewer's concern regarding metric reporting, we have since updated the SPFM pipeline to include component-wise normalization. This ensures that the loss functions ($L_1, L_{1.5}, L_2$) operate on a standardized scale. Specifically for the Chapman dataset, this technique significantly lowers the Sinkhorn distance. We will update the metrics on the final version.
>
> **W3: Train-test distribution shift**
>
> We acknowledge the covariate shift from conditioning the Detail model on generated $x_\text{LF}$ at inference vs. ground-truth $x_\text{LF}$ during training. To mitigate this, we inject Gaussian noise ($\sigma = 0.02 \times \text{std}(x_\text{LF})$) into the conditioning $x_\text{LF}$ during training. This acts as continuous scheduled sampling, making the Detail model robust to minor structural variations in $x_\text{LF}$.
>
> Furthermore, FM models excel at learning LF high-variance manifolds. To quantitatively prove this shift is minimal, the final manuscript will include the Sinkhorn distance between the real $x_\text{LF}$ and generated $x_\text{LF}$ distributions.
>
> **L1: Generalization beyond binary partitions and manual cutoffs**
>
> We appreciate the reviewer highlighting the apparent rigidity in our initial proposed framework. We fully agree that complex signals (like EMG) require finer-grained modeling. To address this, we have generalized SPFM into a multi-band architecture where the frequency cutoffs are analytically derived. Below we briefly introduce the changes:
>
> **Automated, multi-band partitioning**: Rather than relying on an empirical search, the required number of partitions and their cutoff frequencies can be deterministically calculated based on the data's inherent spectral decay rate (the power-law exponent $\beta$) and its frequency span $[f_\text{min}, f_\text{max}]$. This automates hyperparameter selection and naturally extends the framework to $K$ cascaded models for signals with complex mid-frequency bands.
>
> **Intra-dataset variability**: While the frequency partitions are defined at the dataset level, the generative process itself remains dynamic. Patient-specific variations are captured by the LF model and accurately propagated as conditioning information for the HF one, ensuring flexibility.
>
> We hope this addresses your concerns and look forward to your response.

---

> > ### Author Rebuttal · Reviewer_N6Ky · 2026-04-06
> >
> > I thank the authors for the detailed rebuttal. The NTK-based theoretical grounding for gradient democracy (W1), the noise injection strategy for train-test distribution shift (W3), and the extension to automated multi-band partitioning (L1) are constructive additions.
> >
> > However, key evidence remains promised rather than demonstrated — the TSTR experiment, the updated Chapman metrics, and the empirical gradient norm plots are all deferred to the final version. Given that soundness concerns (W1, W3) and evaluation gaps (W2) were central to my assessment, I would need to see these revisions materialized to change my evaluation. I will maintain my original score.

---

> > > ### Author Response · Authors · 2026-04-06
> > >
> > > Thank you for the rebuttal acknowledgement.
> > >
> > > **Regarding the gradient norm plots**: Since the discussion portal does not allow image uploads, we extracted the quantitative evidence for LF gradient dominance on Flow Matching (FM) models trained on domain-agnostic power-law process data (PSD $\propto f^{-\beta}$).
> > >
> > > Throughout training we see that for $\beta = 1$, a LF bin of $\sim 0.2$ Hz has a gradient $L_2$ norm of $\sim 10^{-7}$, while a medium frequency bin of $\sim 5$ Hz has a gradient norm of $\sim 10^{-8}$, and a HF bin of $\sim 40 Hz$ has a gradient norm of $\sim 10^{-9}$. When we analyze $\beta = 2$, for the same bins we obtain gradient $L_2$ norms of $10^{-8}$, $10^{-9}$, and $5^{-9}$ respectively. We hope this serves in lieu of the full gradient norm plots which will be included in the final version.
> > >
> > > **Regarding the TSTR experiment**: We have conducted a downstream classification experiment on the PTB-XL and ISRUC datasets. Following the suggestion by reviewer goUg, we performed linear probing using frozen features extracted from the pre-trained MOMENT time-series foundation model. Using the real and generated data, we extracted the corresponding high-dimensional patch embeddings and trained a logistic regression classifier on three different scenarios: **Real-only** (baseline), **Synthetic-only** (TSTR), and **Mixed** (Real + Synthetic TRSTR). For PTB-XL, we generated a 1:1 synthetic counterpart matching the exact size and class distribution of the real training set, and used the real test data as the evaluation set. For ISRUC, we replicated the size and class distribution of the test set and used the validation data for evaluation.
> > >
> > > The synthetic data was generated using Mamformer with $\mathcal{L}_2$ LF, and $\mathcal{L}_2$ HF losses for PTB-XL, and Mamba, $\mathcal{L}_2$ LF, and $\mathcal{L}_1$ HF losses for ISRUC. These models were taken directly from our ablation study with no fine-tuning, trained for 100K steps and sampled using just 20 Euler integration steps for each model, exactly as reported in the paper.
> > >
> > > **PTB-XL Classification Results**
> > >
> > > | **Scenario** | **Accuracy** | **F1 score** | **AUROC** | **Recall**
> > > | ---------------------- | ---------- |---------- | ---------- | ---------- |
> > > | Real-only (baseline) | **0.91199** | 0.51656 | **0.94450** | 0.51238 |
> > > | Synthetic-only (TSTR) |  0.86908 | 0.47743 | 0.90990 | **0.56899** |
> > > | Mixed (TRSTR) | 0.90931 | **0.53734** | 0.93750 | 0.54576 |
> > >
> > > **ISRUC Classification Results**
> > >
> > > | **Scenario** | **Accuracy** | **F1 score** | **AUROC** | **Recall**
> > > | ---------------------- | ---------- |---------- | ---------- | ---------- |
> > > | Real-only (baseline) | 0.58563 | 0.53336 | 0.83656 | 0.52770 |
> > > | Synthetic-only (TSTR) |  0.58976 | 0.50461 | 0.83959 | 0.51936 |
> > > | Mixed (TRSTR) | **0.61025** | **0.53392** | **0.85120** | **0.54150** |
> > >
> > > When used as a data augmentation technique, our method provides solid improvement in the F1 score and the recall metrics for the PTB-XL dataset, and the accuracy and AUROC metrics are on par with the real-only baseline. The utility of synthetic data is even more pronounced for the ISRUC dataset, where we see improvement across almost all metrics for the synthetic-only (TSTR) scenario, and considerable improvement in all metrics with the mixed (TRSTR) scenario.
> > >
> > > **Regarding the updated Chapman metrics**: As requested, we evaluated the newly normalized SPFM configurations on the Chapman dataset. The table below summarizes the best metrics achieved across the different backbone architectures during our ablation study.
> > >
> > > | **Backbone** | **LF, HF loss** | **Step** | **Sinkhorn** |
> > > | ---------------------- | ---------- | ---------- |---------- |
> > > | Transformer | $L_2 , L_1$ | 25K | 225.28 |
> > > | Mamba |  $L_1 , L_2$ | 50K | 214.86 |
> > > | Mamformer | $L_{1.5} , L_{1.5}$ | 75K | **208.82** |
> > >
> > > We wish to be fully transparent regarding the optimal transport geometry: while the normalization greatly stabilized training, our Sinkhorn distances on the Chapman dataset still trail the baselines.
> > >
> > > Rather than viewing this merely as an empirical limitation, this result serves to validate a core theoretical claim that you have raised in your original review. A simple $K=2$ partition is highly effective for datasets with standard spectral decay. However, Chapman (ECG data) has significant information located in the mid-frequency range. Our theory dictates that the optimal transport geometry for such a signal fundamentally requires a higher number of spectral partitions ($K > 2$) to fully capture the marginal distributions across all relevant frequency bands without approximation gaps. We will openly discuss this limitation in the final manuscript, framing it as a direct validation of the need for future work in automated $K$-scaling.
> > >
> > > We hope that providing these promised empirical results addresses your primary concerns regarding soundness and evaluation, and we kindly ask if you might reconsider your score in light of this new data.

---

### Official Review · Reviewer_ExY2 · 2026-03-13

**Soundness:** 3
**Presentation:** 3
**Significance:** 3
**Originality:** 2
**Overall Recommendation:** 4
**Confidence:** 4

**Summary:**

This paper proposes Spectrally-Partitioned Flow Matching (SPFM), a conditional generative framework for medical time series that splits each signal into low-frequency structure and high-frequency detail using a Butterworth low-pass decomposition, then trains two separate flow-matching models. It addresses the "spectral bias and gradient dominance" problems, where low-frequency (LF) global structures (e.g. heart rhythms) typically drown out high-frequency (HF) local details (e.g., QRS complexes) during training. SPFM decouples signal generation into two specialized, jointly-trained models: a Structure model for LF waveforms and a Detail model for HF components conditioned on the LF structure. The framework is validated across four ECG and EEG datasets.

**Compliance With Llm Reviewing Policy:**

Affirmed.

**Key Questions For Authors:**

* Can you provide direct evidence (e.g., gradient norm statistics for LF vs HF) or controlled comparisons against simpler alternatives such as frequency weighted or multi resolution spectral losses?

* Clarify the exact numerical scheme (Euler/RK4) used at inference. Algorithm 1 currently omits step size and explicit time dependence; how is t integrated into the model during generation?

* Why are multiple SPFM variants shown (each selected for a different metric) while baselines use single configurations? Provide a unified comparison under a consistent model-selection protocol.

* Does the reconstruction of high-frequency components in the super-resolution task improve downstream diagnostic accuracy (e.g. arrhythmia detection) compared to standard bandwidth-extension baselines?

* How does SPFM compare against standard signal-processing or neural bandwidth-extension baselines, and does reconstructing HF components improve any clinically relevant downstream tasks?

**Limitations:**

yes

**Strengths And Weaknesses:**

**Strengths:**

The paper studies whether explicitly separating low- and high-frequency components can improve generative modeling of medical time series. The proposed spectral partitioning framework is intuitive and aligns well with the structure of physiological signals such as ECG and EEG.The proposed Energy Balancing strategy for selecting the spectral cutoff is a reasonable and practical heuristic to mitigate gradient imbalance between frequency components. The empirical evaluation is relatively comprehensive, using multiple complementary metrics (FID, SMSE, Sinkhorn distance) and testing on several ECG and EEG datasets. The paper is generally well structured, with a clear narrative from the spectral bias problem to the proposed SPFM solution and experimental validation.

**Weaknesses:**

* The claim that SPFM mitigates gradient dominance is not directly validated. The paper does not present gradient statistics or controlled comparisons against simpler alternatives such as frequency-weighted losses or multiresolution spectral objectives
* Sampling pseudocode is unclear. The update rule in the algorithm omits explicit time dependence and step-size terms typical of ODE solvers, making the exact numerical integration procedure difficult to reproduce
* The comparison protocol in the ECG experiments is potentially unfair: multiple SPFM variants are reported (each best for a specific metric) while baselines are represented by single configurations
* There are claims of superior performance are somewhat overstated. SPFM improves some metrics but does not consistently outperform baselines across all evaluation measures
* Downstream spectral super-resolution experiment evaluates reconstruction error only and does not demonstrate improvements on clinically relevant downstream tasks (e.g. diagnostic classification)

---

> ### Author Rebuttal · Authors · 2026-03-28
>
> We appreciate the reviewer's positive assessment of our framework. Below we provide responses to the key questions.
>
> **Q1: Direct Evidence for Gradient Democracy**
>
> The reviewer correctly points out the need for formal validation of this claim. To provide theoretical support, we analyze the training dynamics through the lens of Parseval's theorem and the Neural Tangent Kernel (NTK) inverse-variance law presented in [Wang et. al 2025](https://arxiv.org/abs/2503.03206):
>
> **Energy Imbalance (Parseval's theorem)**
>
> Standard flow matching (FM) and diffusion models minimize an $L_2$ (MSE) objective in the time domain. By Parseval's theorem, this is mathematically equivalent to minimizing the energy-weighted error in the frequency domain. For physiological signals with a power-law spectrum $S(f) \propto f^{-\beta}$, the signal energy (and thus the gradient magnitude) is concentrated in the low frequencies. For an ECG signal ($\beta \approx 2$), a component at $50 \text{Hz}$ has 10,000 times less energy than a component at $0.5 \text{Hz}$. Consequently, in a standard FM/diffusion, the high-frequency (HF) gradients are numerically dominated by the (LF) ones.
>
> **NTK inverse-variance law**
>
> This energy disparity is compounded by the NTK inverse-variance which states that the training time $\tau$ required for a specific frequency component $f$ scales as $\tau(f) \propto f^{-\beta}$. Using the same example as above, for a model to learn a $50 \text{Hz}$  component it needs roughly 10,000 times more training steps than it needs to learn a $0.5 \text{Hz}$ one.
>
> Our proposed framework essentially acts like a gradient normalization strategy, which distributes the spectral energy across two models and allows for each to run its own optimization. This rescales the NTK eigenvalues, ensuring that the Detail model receives enough gradient signal to converge within the same training window as the Structure model.
>
> **Q2: Unified Comparison**
>
> To ensure a fair comparison, we plan on updating the result tables with a single configuration of the SPFM framework derived from the ablation study described in section 6.3.
>
> **Q3: Numerical integration and step-size**
>
> We apologize for the omission of time-dependence in Algorithm 1. SPFM uses a first-order Euler scheme where the velocity field $v_\theta$ is conditioned on $t \in [0, 1]$ via sinusoidal embeddings. The update rule should be:
> \begin{equation}
> x_{t_{n+1}} = x_{t_n} + v_\theta(x_{t_n} , t_n, \text{cond}) \cdot \Delta t ,
> \end{equation}
> where $\Delta t = 1/ \text{NFE}$ for the allocated NFE for each model. For the reported experiments, we used $\text{NFE} = 20$ for both the Structure and Detail models, resulting in $\Delta t = 0.05 $.
>
> **Q4: Comparison with frequency-weighted losses**
>
> The reviewer raises a question regarding simpler alternatives like frequency-weighted or multi-resolution losses spectral losses. We argue that while adjusting the loss function can artificially amplify HF gradients during *training*, it fundamentally fails to address the *inference* bottleneck inherent to FM and diffusion models: Even if they are trained until the desired spectral modes are learned, sampling still requires numerical integration (via an ODE solver). HF vector fields are highly oscillatory and numerically stiff. To integrate them without numerical dissipation, the solver requires a very fine step size ($\Delta t \leq \frac{1}{2 f_\text{max}}$). Therefore, a single model would require an impractically large Number of Function Evaluations (NFE) to resolve the full signal capturing the HF tail components.
>
> SPFM addresses this by physically decoupling the generation process. By isolating HF components to a specialized model conditioned on the LF structure, we restrict the vector field to a resolvable numerical regime. Provided each model can successfully resolve its frequency partition, we thus achieve high-fidelity results with a relatively low NFE for each model.
>
> **Q5: Clinical Utility**
>
> We completely agree with the reviewer that evaluating downstream diagnostic performance is a strong measure of practical utility for synthetic medical data. While the short rebuttal window impedes us from running classification experiments, we respectfully argue that our super-resolution experiment provides direct evidence of clinical utility. In clinical electrophysiology, HF components contain critical diagnostic markers. By accurately reconstructing these features, SPFM recovers the markers which diagnostic classifiers, and more importantly, human clinicians, rely on to make accurate diagnostics. We plan on adding a dedicated paragraph and run a final data augmentation experiment in the final manuscript.
>
> We hope we have addressed correctly you questions and look forward to your response.

---

> > ### Author Rebuttal · Reviewer_ExY2 · 2026-04-03
> >
> > Thank you for the response. I have no further comments.

---

> > > ### Author Response · Authors · 2026-04-04
> > >
> > > Thank you for your continued engagement and for confirming the resolution of your concerns. We deeply appreciate the time and effort dedicated to the review of our work, and would like to inform that the constructive feedback during the initial review phase has strengthened the final version of this work.
> > >
> > > If you would like to see the performance of our framework on downstream classifcation, we refer you to the reply rebuttals to reviewers N6Ky, goUg, and Tzed.

---

### Decision · Program_Chairs · 2026-04-30

**Decision:**

Reject

**Comment:**

This paper introduces spectrally partitioned flow matching, a dual-model flow matching framework for medical time series generation. The reviewers found the paper with a clean, well-motivated design and strong computational efficiency.

During the rebuttal and discussion period, however, several reviewers pointed out concerns regarding the theoretical grounding for gradient democracy, empirical gradient norm validation, TSTR classification experiments. Reviewers agreed that the additional experiments during rebuttal were useful to address some of their concerns, but there remains unresolved empirical gaps and performance inconsistencies. Due to the space and time limitation, the new experimental evidence does not revisit the full evaluation suite, which may warrant another round of review. There’s no strong consensus reached for acceptance during the reviewer discussion period.

In light of these feedback, we recommend rejection at this point, and encourage the authors to incorporate the feedback into the revised manuscript.